# Transformer Embeddings of Irregularly Spaced Events and Their Participants

**Chenghao Yang**
Dept. of Computer Science
Columbia University
yangalan1996@gmail.com

**Hongyuan Mei**
Toyota Tech. Institute at Chicago
hongyuan@ttic.edu

**Jason Eisner**
Dept. of Computer Science
Johns Hopkins University
jason@cs.jhu.edu

## Abstract

The neural Hawkes process (Mei & Eisner, 2017) is a generative model of irregularly spaced sequences of discrete events. To handle complex domains with many event types, Mei et al. (2020a) further consider a setting in which each event in the sequence updates a deductive database of facts (via domain-specific pattern-matching rules); future events are then conditioned on the database contents. They show how to convert such a symbolic system into a neuro-symbolic continuous-time generative model, in which each database fact and possible event has a time-varying embedding that is derived from its symbolic provenance.

In this paper, we modify both models, replacing their recurrent LSTM-based architectures with flatter attention-based architectures (Vaswani et al., 2017), which are simpler and more parallelizable. This does not appear to hurt our accuracy, which is comparable to or better than that of the original models as well as (where applicable) previous attention-based methods (Zuo et al., 2020; Zhang et al., 2020a).

## 1 Introduction

It has recently become common to model event sequences by embedding each event into $\mathbb{R}^D$. Event sequences are ubiquitous in real-world applications, such as healthcare, finance, education, commerce, gaming, audio, news, security, and social media. Event embeddings could be used in a variety of downstream applied tasks, similar to word token embeddings in BERT (Devlin et al., 2018).

In this paper, we embed each event using attention over the previous events, using continuous-time positional encodings so as to consider their timing. To build a left-to-right generative model, we also embed possible events at future times in exactly the same way, and use their embeddings to predict their instantaneous probabilities at those times.

Attention-based models (Vaswani et al., 2017) have already become extremely popular for generative modeling of *discrete-time* sequences, such as natural-language documents (Radford et al., 2019; Brown et al., 2020) and proteins (Rao et al., 2021). As we confirm here, they are also effective for modeling sequences that are irregularly spaced in continuous time, even in lower-data regimes.

Our past work on modeling event sequences (Mei & Eisner, 2017; Mei et al., 2019; 2020a;b) used neural architectures based on LSTMs (Hochreiter & Schmidhuber, 1997). That is, predictions at time $t$ were derived from a recurrent encoding of the sequence of timestamped events at times $< t$. However, an attention-based (Transformer-style) architecture has three potential advantages:

① A Transformer does not *summarize* the past. Our predictions at time $t$ can examine an unboundedly large representation of the past (embeddings in $\mathbb{R}^d$ of *every* event before $t$), not merely a fixed-dimensional summary that was computed greedily from left to right (an LSTM's state at time $t$).

② A Transformer's computation graph is broader and shallower. The breadth makes it easier to learn long-distance influences. The shallowness does make it impossible to represent inherently deep concepts such as parity (Hahn, 2020), but it enables greater parallelism: the layer-$\ell$ embeddings can be computed in parallel during training, as they depend only on layer $\ell - 1$ and not on one another.

③ The Transformer architecture is simpler and arguably more natural, while remaining competitive in our experiments. To model the temporal distribution of the next event, all of our models posit embeddings of possible future events that depend on the future event's time $t$. To accomplish this, Mei & Eisner (2017) had to stipulate an arbitrary family of parametric decay functions on $t$, and

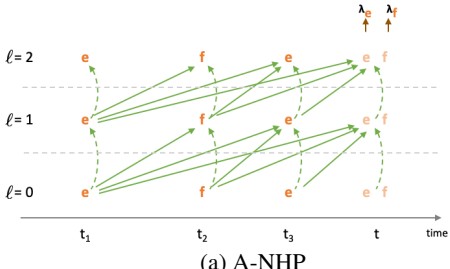
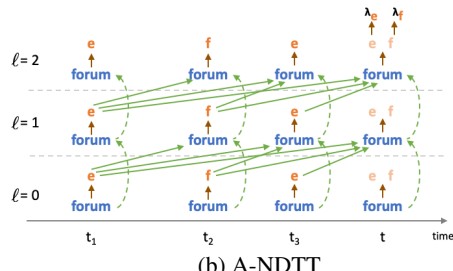

|           |           |
|-----------|-----------|
| (a) A-NHP | (b) A-NDTT |

Figure 1: These figures show how embeddings in the model flow through layers (bottom to top) and through time (left to right). There are two possible event types, $\underline{e}$ and $\underline{f}$, which represent email messages. At the upper right corner of each figure, we obtain their modeled intensities at a certain time $t$, $\lambda_{\underline{e}}(t)$ and $\lambda_{\underline{f}}(t)$, based on the embeddings of the three previous, irregularly spaced observed events. This requires embedding $\underline{e}$ and $\underline{f}$ at time $t$ as if they were observed. If either one actually occurs at time $t$, we will keep its embeddings, which will then affect embeddings of events at times $> t$. Figure (a) shows the basic model of section 3, in which each event's embedding at layer $\ell$ depends ($\longrightarrow$) on all preceding events at layer $\ell - 1$. (The dashed arrows $\dashrightarrow$ reflect residual connections.) Section 4 explains that the $\underline{e} \longrightarrow \underline{f}$ influences can be prevented by dropping the rule $\underline{f} \leftarrow \underline{e}$. Figure (b) shows an A-NDTT model from section 5, in which the company forum's embedding at layer $\ell$ depends ($\longrightarrow$) on all preceding events at layer $\ell - 1$ (via $\leftarrow$ rules). The events or possible events at layer $\ell$ do *not* depend directly on preceding events; instead, their embeddings at time $t$ are derived ($\longrightarrow$) from the forum's embedding at time $t$ (via $:-$ rules).

the neuro-symbolic framework of Mei et al. (2020a) required a complex method for pooling the parameters of these decay functions. But in our present method, no decay functions are required to allow embeddings and probabilities to drift over time. The embeddings are constructed "from scratch" at each time $t$ simply by attending to the set of previous events, using $t$-specific query vectors that include a continuous positional embedding of $t$. As $t$ increases, the attention weights over the previous events vary continuously with $t$, so the embeddings and probabilities do so as well.

We present a series of increasingly sophisticated methods. Section 2 explains how to embed events in context (like continuous-time BERT). Section 3 turns this into a generative point process model that can predict the time and type of the next event (like continuous-time GPT). In section 4, we allow a domain expert to write simple rules that control attention, constraining which events can "see" which previous events and with what parameters. Finally, section 5 allows the domain expert to write more complex rules, using our previously published **Datalog through time** formalism (Mei et al., 2020a). These rules allow events to interact with a symbolic deductive database that tracks facts over time so that the neural architecture does not have to learn how to do so. As in Mei et al. (2020a), we define time-varying embeddings for all facts in the database and all events that are possible given those facts, using parameters associated with the rules that established the facts and possible events.

In the end, we arrive at attention-based versions of the NHP (Mei & Eisner, 2017) and NDTT (Mei et al., 2020a) frameworks, which we refer to as A-NHP (section 3) and A-NDTT (section 5). We evaluate them in section 7, showing comparable or better accuracy. We release our code.

## 2 CONTINUOUS-TIME TRANSFORMER FOR EMBEDDING EVENTS

Suppose we observe $I$ events over a fixed time interval $[0, T]$. Each event is denoted mnemonically as $e@t$ (i.e., "type $e$ *at* time $t$"), where $e \in \mathcal{E}$ is the type of event (drawn from a finite set $\mathcal{E}$). The observed **event sequence** is $e_1@t_1, e_2@t_2, \ldots, e_I@t_I$, where $0 < t_1 < t_2 < \ldots < t_I < T$. For any event $e@t$, we can compute an **embedding** $[\![e]\!](t) \in \mathbb{R}^D$ by attending to its **history** $\mathcal{H}(e@t)$—a set of *relevant* events. (For the moment, imagine that $\mathcal{H}(e@t)$ consists of *all* the observed events $e_i@t_i$.) More precisely, $[\![e]\!](t)$ is the concatenation of layer-wise embeddings $[\![e]\!]^{(0)}(t), [\![e]\!]^{(1)}(t), \ldots, [\![e]\!]^{(L)}(t)$. For $\ell > 0$, the layer-$\ell$ embedding of $e@t$ is computed as

$$[\![e]\!]^{(\ell)}(t) \overset{\text{def}}{=} \underbrace{[\![e]\!]^{(\ell-1)}(t)}_{\text{residual connection}} + \tanh\left( \sum_{f@s \in \mathcal{H}(e@t)} \frac{\mathbf{v}^{(\ell)}(f@s)\, \alpha^{(\ell)}(f@s, e@t)}{1 + \sum_{f@s \in \mathcal{H}(e@t)} \alpha^{(\ell)}(f@s, e@t)} \right) \quad (1)$$

where the unnormalized attention weight on each relevant event $f@s \in \mathcal{H}(k@t)$ is

$$\alpha^{(\ell)}(f@s, e@t) \overset{\text{def}}{=} \exp\left( \frac{1}{\sqrt{D}} \mathbf{k}^{(\ell)}(f@s)^{\top} \mathbf{q}^{(\ell)}(e@t) \right) \in \mathbb{R} \quad (2)$$

In layer $\ell$, $\mathbf{v}^{(\ell)}$, $\mathbf{k}^{(\ell)}$, and $\mathbf{q}^{(\ell)}$ are known as the **value**, **key**, and **query** vectors and are extracted from the layer-$(\ell-1)$ event embeddings using learned layer-specific matrices $\mathbf{V}^{(\ell)}, \mathbf{K}^{(\ell)}, \mathbf{Q}^{(\ell)}$:

$$\mathbf{v}^{(\ell)}(e@t) \stackrel{\text{def}}{=} \mathbf{V}^{(\ell)} \left[1; [\![t]\!]; [\![e]\!]^{(\ell-1)}(t)\right] \quad \mathbf{k}^{(\ell)}(e@t) \stackrel{\text{def}}{=} \mathbf{K}^{(\ell)} \left[1; [\![t]\!]; [\![e]\!]^{(\ell-1)}(t)\right] \quad \mathbf{q}^{(\ell)}(e@t) \stackrel{\text{def}}{=} \mathbf{Q}^{(\ell)} \left[1; [\![t]\!]; [\![e]\!]^{(\ell-1)}(t)\right]$$

$$\text{(3a)} \qquad\qquad\qquad\qquad \text{(3b)} \qquad\qquad\qquad\qquad \text{(3c)}$$

As the base case, $[\![e]\!]^{(0)}(t) \stackrel{\text{def}}{=} [\![e]\!]^{(0)}$ is a learned embedding of the event type $e$. $[\![t]\!]$ denotes an embedding of the time $t$. We cannot learn absolute embeddings for all real numbers, so we fix

$$[\![t]\!]_d = \sin(t/(m \cdot (\tfrac{5M}{m})^{\frac{d}{D}})) \text{ if } d \text{ is even} \qquad [\![t]\!]_d = \cos(t/(m \cdot (\tfrac{5M}{m})^{\frac{d-1}{D}})) \text{ if } d \text{ is odd} \quad (4)$$

where $0 \le d < D$ are the dimensions and our choices of $m, M$ are explained in Appendix A.

Crucially, to compute the layer-$\ell$ embedding of an event, equations (1)–(3) need only the layer-$(\ell-1)$ embeddings of the relevant events in its history. This lets us compute the layer-$\ell$ embeddings of all events in parallel. Note that equations (1)–(3) are simplifications of the traditional Transformer, since this ablation performed equally well in our pilot experiments (see Appendix A).

The set of relevant events $\mathcal{H}(e@t)$ could be defined in a task-specific way. For example, to pretrain BERT-like embeddings (Devlin et al., 2018), we might use a corrupted version of $\{e_1@t_1, \ldots, e_I@t_I\}$ in which some $e_i@t_i$ have been removed or replaced with mask$@t_i$. Such embeddings could be pretrained with a BERT-like objective and then fine-tuned to predict properties of the observed events.

## 3 GENERATIVE MODELING OF CONTINUOUS-TIME EVENT SEQUENCES

In this paper, we focus on the task of predicting future events given past ones. At any time $t$, we would like to know what will happen at that time, given the actual events that happened *before* $t$. Our generative model is analogous to a Transformer language model (Radford et al., 2019; Brown et al., 2020), which, at each time $t \in \mathbb{N}$, defines a probability distribution over the words in the vocabulary.

In our setting, however, $t \in \mathbb{R}$. With probability 1, *nothing* happens at time $t$. Each possible event $e$ in our vocabulary has only an *infinitesimal* probability of occurring at time $t$. We write this probability as $\lambda_e(t)dt$ where $\lambda_e(t) \in \mathbb{R}^+$ is called the (Poisson) **intensity** of type-$e$ events at time $t$. More formally, the probability of such an event occurring during $[t, t + \epsilon)$ approaches $\lambda_e(t)\epsilon$ as $\epsilon \to^+ 0$.

Thus, our modeling task is to model $\lambda_e(t)$ (as in, e.g., Hawkes, 1971; Du et al., 2016; Mei & Eisner, 2017). We model $\lambda_e(t)$ as a function of the top-layer embedding of the *possible* event $e@t$:

$$\lambda_e(t) \stackrel{\text{def}}{=} \text{softplus}(\mathbf{w}_e^\top[1; [\![e]\!]^L(t)], \tau_e) \quad \text{where softplus}(x, \tau) = \tau \log(1 + \exp(x/\tau)) > 0 \quad (5)$$

with learnable parameters $\mathbf{w}_e$ and $\tau_e > 0$. We do this separately for each possible $e@t$, computing the embedding $[\![e]\!]^L(t)$ using equations (1)–(3). The softplus transfer function is inherited from the neural Hawkes process (Mei & Eisner, 2017). To ensure that our model is generative, we compute $[\![e]\!](t)$ from only *previous* events. That is, $\mathcal{H}(e@t)$ in equation (1) may contain any or all of the previously generated events $e_i@t_i$ for $t_i < t$, but it may not contain any for which $t_i > t$. We call this model the **attentive Neural Hawkes process**, or **A-NHP**, and evaluate it in section 7.

Our model's log-likelihood has the same form as for any autoregressive multivariate point process:

$$\sum_{i=1}^{I} \log \lambda_{e_i}(t_i) - \int_{t=0}^{T} \sum_{e=1}^{E} \lambda_e(t)dt \quad (6)$$

Derivations of this formula can be found in previous work (e.g., Hawkes, 1971; Liniger, 2009; Mei & Eisner, 2017). We can estimate the parameters by locally maximizing the log-likelihood (6) by any stochastic gradient method. Intuitively, each $\log \lambda_{e_i}(t_i)$ is increased to explain why the observed event $e_i$ happened at time $t_i$, while $\int_{t=0}^{T} \sum_{e=1}^{E} \lambda_e(t)dt$ is decreased to explain why no event of any possible type $e \in \{1, \ldots, E\}$ ever happened at other times.

Appendix D gives training details, including Monte Carlo approximations to the integral in equation (6), as well as noting alternative training objectives. Given the learned parameters, we may wish to sample from the model given the past history, or make a minimum Bayes risk prediction about the next event. Recipes can be found in Appendix E.

Notice that equation (5) is rather expensive compared to previous work, since it computes a deep embedding of the possible event $e@t$ just for the purpose of finding its intensity—and the algorithms of Appendices D–E require computing the intensities of *many* possible events. Appendix A offers a speedup that shares embeddings among similar events, but it also explains why different events may sometimes have to be embedded differently to support the selective attention in sections 4–5 below.

## 4 MULTI-HEAD SELECTIVE ATTENTION

We now present a simple initial version of *selective* attention. As in a graphical model, not all events should be able to influence one another directly. Consider a scenario with two event types: $\underline{e}$ means that Eve emails Adam, while $\underline{f}$ means that Frank emails Eve. As Frank does not know when Eve emailed Adam, past events of type $\underline{e}$ cannot influence his behavior. Therefore, $\mathcal{H}(\underline{f}@t)$ should include past events of type $\underline{f}$ but not $\underline{e}$, so that the embedding of $\underline{f}@t$ and hence the intensity function $\lambda_{\underline{f}}(t)$ pay zero attention to $\underline{e}$ events. In contrast, $\mathcal{H}(\underline{e}@t)$ should still include past events of both types, since both are visible to Eve and can influence her behavior.

We describe this situation with the edges $\underline{f} \leftarrow \underline{f}$, $\underline{e} \leftarrow \underline{e}$, $\underline{e} \leftarrow \underline{f}$. These are akin to the edges in a directed graphical model. They specify the sparsity pattern of the **influence matrix** (or **Granger causality matrix**) that describes which past events can influence which future events. There is a long history of estimating this matrix from observed sequence data (e.g., Xu et al., 2016; Zhang et al., 2021), even with neural influence models (Zhang et al., 2020b). In the present paper, however, we do not attempt to estimate this sparsity pattern, but assume it is provided by a human domain expert. Incorporating such domain knowledge into the model can reduce the amount of training data that is needed. Edges like $\underline{e} \leftarrow \underline{f}$ can be regarded as simple cases of the NDTT **rules** in section 5 below.

Such rules also affect how we apply attention. When Eve decides whether to email Adam ($\underline{e}@t$), we may reasonably suppose that she *separately* considers the embeddings of the past $\underline{e}$ events (e.g., "when were my last relevant emails *to* Adam?") versus the past $\underline{f}$ events (e.g., "what have I heard recently *from* Frank?"). Hence, we associate different attention heads with the two rules that affect $\underline{e}$, namely $\underline{e} \leftarrow \underline{e}$ and $\underline{e} \leftarrow \underline{f}$. These heads may have different parameters, so that they seek out and obtain different information from the past via different queries, keys, and values. In general, we replace equation (1) with

$$\llbracket e \rrbracket^{(\ell)}(t) \overset{\text{def}}{=} \llbracket e \rrbracket^{(\ell-1)}(t) + \tanh \left( \sum_r \boxed{e}_r^{(\ell)}(t) \right) \tag{7}$$

$$\boxed{e}_r^{(\ell)}(t) \overset{\text{def}}{=} \sum_{f@s \in \mathcal{H}_r(e@t)} \frac{\mathbf{v}_r^{(\ell)}(f@s)\, \alpha_r^{(\ell)}(f@s, e@t)}{1 + \sum_{f@s \in \mathcal{H}_r(e@t)} \alpha_r^{(\ell)}(f@s, e@t)} \tag{8}$$

where $r$ in the summation ranges over rules $e \leftarrow \cdots$. The history $\mathcal{H}_r(e@t)$ contains only those past events $f@s$ that rule $r$ makes visible to $e$. If there are no such events, or they have small attention weights (are only weakly relevant to $e@t$ as discussed in Appendix A), then rule $r$ will contribute little or nothing to the sum in equation (7). The attention weights $\alpha_r$ and vectors $\mathbf{v}_r$ are defined using versions of equations (2)–(3) with $r$-specific parameters.[5]

In short, each rule looks at the context separately, through its own attention weights determined by its own parameters. The rule already specifies symbolically which past events can get nonzero attention in the first place, so it makes sense for the rule to also provide the parameters that determine the attention weights and value projections. Further discussion is given in Appendix A.

## 5 ATTENTIVE NEURAL DATALOG THROUGH TIME

Edges such as $\underline{e} \leftarrow \underline{f}$ can be regarded as simple examples of rules in an NDTT program (Mei et al., 2020a, section 2). We briefly review this formalism and then extend our approach from section 4 to handle all NDTT programs.

A **Datalog through time (DTT)** program describes *possible* sequences of events, much as a regular expression describes legal sequences of characters. A DTT program for a particular domain specifies how each event automatically updates a **database**, adding or removing facts. In this way, the past events $e_1, \ldots, e_i$ sequentially construct a database. This database then determines which event types (if any) can happen next: the next event $e_{i+1}$ can be $\underline{f}$ only if $\underline{f}$ is currently a fact in the database.

Thus, an event may appear in the database as a fact, meaning that the event is possible. We use variables $e, f$ to range over events, but variables $g, h$ to range over any facts (both events and non-events). Literal examples of facts are shown in orange if they are events (e.g., $\underline{f}$), and in blue otherwise.

A **neural Datalog through time (NDTT)** program is a DTT program augmented with some dimensionality declarations (Appendix C). A rule that adds a fact to the database now also computes a vector embedding of that fact, or updates the existing embedding if the fact was already in the database.

Notice that the dimensionality of the embedded database changes as the database grows and shrinks over time. Nonetheless, the model has a fixed number of parameters associated with the fixed set of rules of the NDTT program. As we will see, rules can contain variables, allowing a small set of rules to model a large set of event types (i.e., parameter sharing).

If $\underline{\mathtt{f}}$ is a fact in the database at time $t$, meaning that event $\underline{\mathtt{f}}$ is possible at time $t$, then its embedding $[\![\underline{\mathtt{f}}]\!]^L(t)$ determines its intensity $\lambda_{\underline{\mathtt{f}}}(t)$ via equation (5), as before. Thus, where a DTT program only describes which event sequences are *possible*, an NDTT program also describes how *probable* they are.

Although the *set* of database facts is modified only when an event occurs, the facts' *embeddings* are time-sensitive and evolve as the events that added them to the database recede into the past. This allows event intensities such as $\lambda_{\underline{\mathtt{f}}}(t)$ to wax and wane continuously as time elapses.

**Datalog.** We now give details. We begin with Datalog (Ceri et al., 1989), a traditional formalism for deductive databases. A deductive database holds both **extensional facts**, which are placed there by some external process, and **intensional facts**, which are transitively deduced from the extensional facts. A Datalog program is simply a set of rules that govern these deductions:

- $h$ `:-` $g_1$`,` `...,` $g_n$ says to **deduce** $h$ at any time $t$ when $g_1, \ldots, g_n$ are all true (in the database).

A single rule can license many deductions. That is because the facts can have structured names, and $h, g_1, \ldots g_n$ can be patterns that match against those names, using capitalized identifiers as variables. A model of filesystem properties might have a rule like `open`(U,D) `:-` `user`(U)`,` `group`(G)`,` `document`(D)`,` `member`(U,G)`,` `readable`(D,G). In English, this says that U can open D at any time when user U is a member of some group G such that document D is readable by G.

**Datalog through time.** Whenever extensional facts are added or removed, the intensional facts are instantly recomputed according to the deductive rules. DTT is an extension in which extensional facts are *automatically* added and removed when the database is notified of the occurrence of events. This behavior is governed by two additional rule types:

- $h$ `<-` $f$`,` $g_1$`,` `...,` $g_n$ says to **add** $h$ at any time $s$ when event $f$ occurs and the $g_i$ are all true.
- $!h$ `<-` $f$`,` $g_1$`,` `...,` $g_n$ says to **remove** $h$ at any time $s$ satisfying the same conditions.

Thus, the proposition $h$ is true at time $t$ (i.e., appears as a fact in the database at time $t$) iff either ❶ $h$ is deduced at time $t$, or ❷ $h$ was added at some time $s < t$ and never removed at any time in $(s, t)$.

In our previous example, `editing`(U,D) `<-` `open`(U,D)`,` `member`(U,G)`,` `writeable`(U,G) records in the database that user U is editing D, once they have opened it with appropriate permissions. (As a result, edit events might become possible via a deductive rule `edit`(U,D) `:-` `editing`(U,D).)

**Neural Datalog through time.** It would be difficult to train a neural architecture to encode thousands or millions of structured boolean facts about the world in its state and to systematically keep those facts up to date in response to possibly rare events. As a *neuro-symbolic* method, NDTT delegates that task to a symbolic database governed by manually specified DTT rules. However, it also augments the database: iff a proposition $h$ appears as a fact in the NDTT database at time $t$, it will be associated not only with the simple truth value `true` but also with an embedding $[\![h]\!](t)$. This embedding is a *learned representation* of that fact at that time, and can be trained to be useful in downstream tasks. It captures details of when and how that fact was established (the fact's **provenance**), since it is computed using learned parameters associated with the rules that deduced and/or added it.

For example, a user's embedding might be constructed using attention over all the past events that have affected the user, via rules of the form `user`(U) `<-` $\cdots$. This summarizes the user's state. Similarly, a document's embedding might be constructed using attention over all the edits to it, considering the editing user's state at the time of the edit: `document`(D) `<-` `edit(U,D)`, `user`(U).

**Embeddings from NDTT rules.** Our goal is to provide new formulas for the embeddings $[\![h]\!](t)$, based on Transformer-style attention rather than LSTM-style recurrence. We call this **attentive NDTT**, or **A-NDTT**. This gives a new way to map an NDTT program to a neural architecture. The potential advantages for accuracy, efficiency, and simplicity were explained in section 1.

Intuitively, the `<-` rules will govern the "horizontal" flow of information through time (by defining attentional connections as we saw in section 4), while the `:-` rules will govern the "vertical" flow of

information at a given time (by defining feed-forward connections). These are, of course, the two major mechanisms in Transformer architectures.

Under A-NDTT, the layer-$\ell$ embedding of $h@t$ is

$$\llbracket h \rrbracket^{(\ell)}(t) \stackrel{\text{def}}{=} \llbracket h \rrbracket^{(\ell-1)}(t) + \tanh\left([h]^{(\ell)}(t) + \sum_r \boxed{h}_r^{(\ell)}(t)\right) \tag{9}$$

which is an augmented version of equation (7). Suppose $h$ is true at time $t$ because it was added by rule $r$ (i.e., condition ❷). Then the summand $\boxed{h}_r^{(\ell)}(t)$ exists and is computed much as in equation (8), now with attention over all "add times" $s$. In other words, $\mathcal{H}_r(h@t)$ in equation (8) includes just those past events $f@s$ such that $f$ added $h$ via $r$ at some time $s < t$ and $h$ was never removed at any time in $(s, t)$.

More precisely, when the rule $h \leftarrow f, g_1, \ldots, g_n$ causes $h@t$ to attend to the specific past event $f@s$, we actually want attention to consider the embedding at time $s$ not just of $f$, but of the *entire* add condition $f, g_1, \ldots, g_n$. Thus, we replace $f@s$ with $(f, g_1, \ldots, g_n)@s$ throughout equation (8). The attention key of this add condition is defined as $\mathbf{k}^{(\ell)}((f, g_1, \ldots, g_n)@s) \stackrel{\text{def}}{=} \mathbf{K}_r^{(\ell)}\left[1; \llbracket s \rrbracket; \llbracket f \rrbracket^{(\ell-1)}(s); \llbracket g_1 \rrbracket^{(\ell-1)}(s); \ldots \llbracket g_n \rrbracket^{(\ell-1)}(s)\right]$ (compare equation (3b)). Its attention value $\mathbf{v}^{(\ell)}((f, g_1, \ldots, g_n)@s)$ is defined analogously, using a different matrix $\mathbf{V}_r^{(\ell)}$.

The above handles the $\leftarrow$ rules. As for the $:-$ rules, the vector $[h]^{(\ell)}(t)$ in equation (9) sums over all the ways that $h$ can be deduced at time $t$ (i.e., condition ❶). This does not involve attention, so we exactly follow Mei et al. (2020a, equations (3)–(6)):

$$[h]^{(\ell)}(t) = \sum_r \bigoplus_{g_1, \ldots, g_n}^{\beta_r} \mathbf{W}_r[1; \llbracket g_1 \rrbracket(t); \ldots; \llbracket g_n \rrbracket(t)] \tag{10}$$

where $r$ ranges over rules, and $(g_1, \ldots, g_n)$ ranges over all tuples of facts at time $t$ such that $h :- g_1, \ldots g_n$ matches rule $r$ (and thus deduces $h$ at time $t$). The operator $\bigoplus^{\beta_r}$ is a softmax-pooling operator with a learned inverse temperature $\beta_r$. If $h$ is not deduced at time $t$ by any instantiation of $r$, then $r$ has no effect on the sum (10), since pooling the empty set with $\bigoplus^{\beta_r}$ returns $\mathbf{0}$.

**Example.** Mei et al. (2020a) give many examples of NDTT programs. Here is a simple example to illustrate the use of $:-$ rules. $\underline{e}$ means that Eve posts a message to the company forum, while $\underline{f}$ means that Frank does so. Once the forum is created by a $\underline{\texttt{create}}$ event, its existence is a fact (called $\texttt{forum}$) whose embedding (called $\llbracket \texttt{forum} \rrbracket$) always reflects all messages posted so far to the forum. Until the forum is destroyed, Eve and Frank can post to it, and the embeddings and intensities of their messages depend on the current state of the forum:

```
1  forum ← create.          3  forum ← e.          5  e :- forum.
2  !forum ← destroy.         4  forum ← f.          6  f :- forum.
```

The resulting neural architecture is drawn in Figure 1b. If the company grows from 2 to $K$ employees, then the program needs $O(K)$ rules and hence $O(K)$ parameters, which define how each employee's messages affect the forum and vice-versa. Without the $:-$ rules, we would have to list out $O(K^2)$ rules such as $\underline{e} \leftarrow \underline{f}$ and hence would need $O(K^2)$ parameters, which define how each employee's messages affect every other employee's messages directly; this case is drawn in Figure 1a.

Appendix B and Figure 4 spell out an enhanced version of this example that makes use of variables, so that all $K$ employees can be governed by a constant ($O(1)$) number of rules.

**Discussion.** NDTT rules enrich the notion of "influence matrix" from section 4. Events traditionally influence the intensities of subsequent events, but NDTT $\leftarrow$ rules more generally let them influence the *embeddings* of subsequent *facts* (and hence the intensities of any events among those facts). Furthemore, NDTT $:-$ rules let facts influence the embeddings of *contemporaneous* facts.

Each $\leftarrow$ rule $r$ can be seen as defining the fixed sparsity pattern of a large influence matrix, along with parameters for computing its nonzero entries from context at each attention layer. The size of this matrix is determined by the number of ways to instantiate the variables in the rule. The entries of the matrix are normalized versions of the attention weights $\alpha_r$. The influences of different $\leftarrow$ rules $r$ are combined by equation (9) and are modulated by nonlinearities.

Overall, (A-)NDTT models learn representations, much like pretrained language models (Peters et al., 2018; Radford et al., 2019). They learn continuous embeddings of the facts in a discrete database, using a neural architecture that is derived from the symbolic rules that deduce these facts and update them

in response to events. The facts change at discrete times but their embeddings change continuously. We train the model so that the embeddings of possible events predict how likely they are to occur.

## 6 RELATED WORK

Multivariate point processes have been widely used in real-world applications, including document stream modeling (He et al., 2015; Du et al., 2015a), learning Granger causality (Xu et al., 2016; Zhang et al., 2020b; 2021), network analysis (Choi et al., 2015; Etesami et al., 2016), recommendation systems (Du et al., 2015b), and social network analysis (Guo et al., 2015; Lukasik et al., 2016).

Over the recent years, various neural models have been proposed to expand the expressiveness of point processes. They mostly use recurrent neural networks, or LSTMs (Hochreiter & Schmidhuber, 1997): in particular Du et al. (2016); Mei & Eisner (2017); Xiao et al. (2017a;b); Omi et al. (2019); Shchur et al. (2020); Mei et al. (2020a); Boyd et al. (2020). Models of this kind enjoy continuous and infinite state spaces, as well as flexible transition functions, thus achieving superior performance on many real-world datasets, compared to classical models such as the Hawkes process (Hawkes, 1971).

The **Transformer Hawkes process** (Zuo et al., 2020) and **self-attentive Hawkes process** (Zhang et al., 2020a) were the first papers to adapt generative Transformers (Vaswani et al., 2017; Radford et al., 2019; Brown et al., 2020) to point processes. The Transformer architecture allows their models to enjoy unboundedly large representations of histories, as well as great parallelism during training (see ① and ② in section 1). As section 3 discussed, both models—as well as subsequent attention-based models (Enguehard et al., 2020; Sharma et al., 2021)—derive the intensity $\lambda_e(t)$ from $[\![f]\!](s)$ where $f@s$ is the latest *actual* event before $t$. (The THP takes $\lambda_e(t)$ to be a softplus function of $\mathbf{w}_e^\top [1; t/s; [\![f]\!](s)]$. The SAHP defines $\lambda_e(\cdot)$ as a function that exponentially decays toward an asymptote, computing the 3 parameters of this function from $e$ and $[\![f]\!](s)$.) In contrast (see ③ in section 1), our model derives $\lambda_e(t)$ from $[\![e]\!]t$—the embedding of the *possible* event $e@t$, which is computed using $e$- and $t$-specific attention over all past events. Zhu et al. (2021, section 3.1) independently proposed this approach but did not evaluate it experimentally.

## 7 EXPERIMENTS

On several synthetic and real-world datasets, we evaluate our model's held-out log-likelihood, and its success at predicting the time and type of the next event. We compare with multiple strong competitors. Experimental details not given in this section can be found in Appendix F.

We implemented our A-NDTT framework using PyTorch (Paszke et al., 2017) and pyDatalog (Carbonell et al., 2016), borrowing substantially from the public implementation of NDTT (Mei et al., 2020a). We also built a faster, GPU-friendly PyTorch implementation of our more restricted A-NHP model (see section 7.1 below). Our code and datasets are available at `https://github.com/yangalan123/anhp-andtt`.

For the competing models, we made use of their published implementations.[1] References and URLs are provided in Appendix F.2.

### 7.1 COMPARISON OF DIFFERENT TRANSFORMER ARCHITECTURES

We first verify that our continuous-time Transformer is competitive with three state-of-the-art neural event models. The four models we compare are

**Transformer Hawkes Process (THP)** (Zuo et al., 2020). See section 6.

**Self-Attentive Hawkes Process (SAHP)** (Zhang et al., 2020a). See section 6.

**Neural Hawkes Process (NHP)** (Mei & Eisner, 2017). This is not an attention-based model. At any time $t$, NHP uses a continuous-time LSTM to summarize the events over $[0, t)$ into a multi-dimensional state vector, and conditions the intensities $\lambda_e(t)$ of all event types on that state.

**Attentive Neural Hawkes Process (A-NHP)** This is our unstructured generative model from section 3. Since this model does not use selective attention, we speed up the intensity computations by defining them in terms of a single coarse event type, as described in Appendix A. Thus, each event intensity

---

[1]On some datasets, our replicated results are different from their papers. We confirmed that our results are correct via personal communication with the lead authors of Zhang et al. (2020a) and Zuo et al. (2020).

$\lambda_e(t)$ is computed by attention over all previous events, where the attention weights are independent of $e$. This parameter-sharing mechanism resembles the NHP, except that we now use a Transformer in place of an LSTM.

In a pilot experiment, we drew sequences from randomly initialized models of all 4 types (details in Appendix F.1.1), and then fit all 4 models on each of these 4 synthetic datasets. We find that NHP, SAHP, and A-NHP have very close performance on all 4 datasets (outperforming THP, especially at predicting timing, except perhaps on the THP dataset itself); see Figure 5 in Appendix F.1.1 for results. Thus, A-NHP is still a satisfactory choice even when it is misspecified. This result is reassuring because A-NHP has less capacity in some ways (the circuit depth of a Transformer is fixed, whereas the circuit depth of an LSTM grows with the length of the sequence) and excess capacity in other ways (the Transformer has unbounded memory whereas the LSTM has finite-dimensional memory).

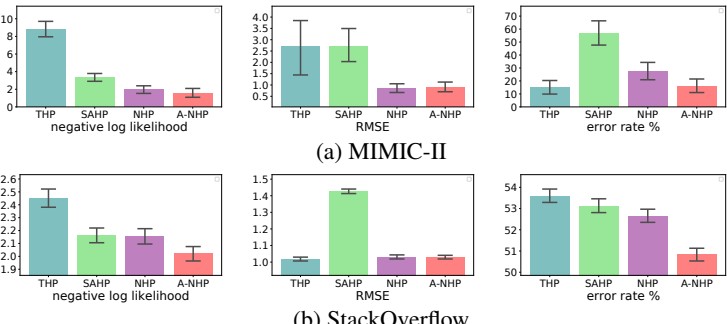

(a) MIMIC-II

(b) StackOverflow

Figure 2: Evaluation results (smaller is better) with 95% bootstrap confidence intervals[6] on the two real-world datasets, comparing THP, SAHP, and NHP with our A-NHP model. RMSE evaluates the predicted time of the next event (root mean squared error), while error rate evaluates its predicted type given its time.

We then fit all 4 models to the following two benchmark real-world datasets.[2]

**MIMIC-II** (Lee et al., 2011). This dataset is a collection of de-identified clinical visit records of Intensive Care Unit patients for 7 years.Each patient has a sequence of hospital visit events, and each event records its time stamp and disease diagnosis.

**StackOverflow** (Leskovec & Krevl, 2014). This dataset represents two years of user awards on a question-answering website: each user received a sequence of badges (of 22 different types).

On MIMIC-II data (Figure 2a), our A-NHP is always a co-winner on each of these tasks; but the other co-winner (NHP or THP) varies across tasks. On StackOverflow data (Figure 2b), our A-NHP is clearly a winner on 2 out of 3 tasks and is tied with NHP on the third. Compared to NHP, A-NHP also enjoys a computational advantage, as discussed in sections 1 and 2. Empirically, training an A-NHP only took a fraction of the time that was needed to train an NHP, when sequences are reasonably long. Details can be found in Table 2 of Appendix F.3.

## 7.2 A-NDTT VS. NDTT

Now we turn to the structured modeling approach presented in section 5. We compare A-NDTT with NDTT on the RoboCup dataset and IPTV dataset proposed by Mei et al. (2020a). In both cases, we used the NDTT program written by Mei et al. (2020a). The rules are unchanged; the only difference is that our A-NDTT has the new continuous-time Transformer in lieu of the LSTM architecture. We also evaluated the unstructured NHP and A-NHP models on these datasets.

**RoboCup** (Chen & Mooney, 2008). This dataset logs the actions (e.g., `kick`, `pass`) of robot soccer players in the RoboCup Finals 2001–2004. The ball is frequently transferred between players (by passing or stealing), and this dynamically changes the set of possible event types (e.g., only the ball possessor can kick or pass). There are $K = 528$ event types over all time, but only about 20 of them are possible at any given time. For each prefix of each held-out event sequence, we used minimum Bayes risk to predict the next event's time, and to predict its participant(s) given its time and action type.

---

[1]If we are not taking enough Monte Carlo samples to get a stable estimate of log-likelihood, then this will appropriately be reflected in wider error bars. This is because when computing a bootstrap replicate, we recompute our Monte Carlo estimate of the log-likelihood of each sequence. Hence, our bootstrap confidence intervals take care to include the variance due to the stochastic evaluation metric. For the Monte Carlo settings we actually used (Appendix D), this amounts to about 1% of the width of the error bars.

[2]For these datasets, we used the preprocessed versions provided by Mei & Eisner (2017). More details about them can be found in Appendix F.1.2.

**IPTV** (Xu et al., 2018). This dataset contains records of 1000 users watching 49 TV programs over the first 11 months of 2012. Each event has the form `watch`(U,P). Given each prefix of the test event sequence, we attempted to predict the next test event's time $t$, and to predict its program P given its actual time $t$ and user U.

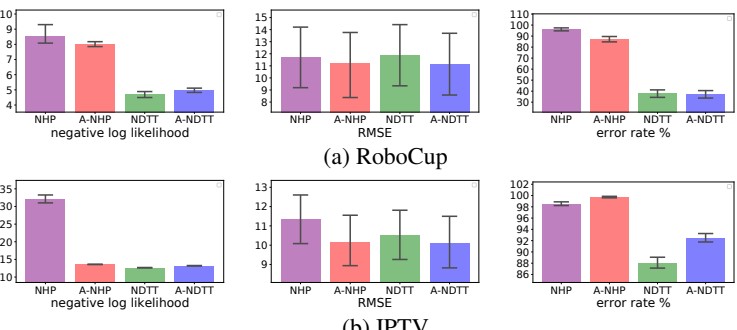

(a) RoboCup

(b) IPTV

Figure 3: Evaluation results with 95% bootstrap confidence intervals[6] on the RoboCup and IPTV datasets. Evaluation methods are the same as in Figure 2. Note that the training objective was log-likelihood.

The Robocup results are shown in Figure 3a. As in section 7.1, we find that A-NHP performs better than NHP on all the evaluation metrics; on log-likelihood and event type prediction, A-NHP is significantly better (paired permutation test, $p < 0.05$). We now inject domain knowledge into both the LSTM and Transformer approaches, by deriving architectures based on the RoboCup NDTT program (which specifies, for example, that only the ball possessor can kick or pass). The resulting models—NDTT and A-NDTT—are substantial and significant improvements, considerably reducing both the log-likelihood and the very high error rate on event type prediction. NDTT and A-NDTT are not significantly different from each other: since NDTT already knows which past events might be relevant, perhaps it is not sorely in need of the Transformer's ability to scan an unbounded history for relevant events.[3] Appendix F.5 includes more results of A-NDTT vs. NDTT broken down by action types.

Additionally, while A-NDTT does not improve the overall accuracy for this particular NDTT program and dataset, it does achieve overall comparable accuracy with a *simpler* and *shallower* architecture (②–③ in section 1). Like other Transformers, the A-NDTT architecture could be trained on a GPU with parallelism, as outlined in Appendix F.4 (future work).

The IPTV results are shown in Figure 3b. In this case, the log-likelihood of NHP can be substantially and significantly improved either by using rules (as for Robocup) or by using attention, or both. The error rate on predicting the next event type is again very high for NHP, and is substantially and significantly reduced by using rules—although not as much under the A-NDTT architecture as under the original NDTT architecture.

## 8 CONCLUSION

We showed how to generalize the Transformer architecture to sequences of discrete events in continuous time. Our architecture builds up rich embeddings of actual and possible events at any time $t$, from lower-level representations of those events and their contexts. We usually train the model so that the embedding of a possible event predicts its intensity, yielding a flexible generative model that supports parallel computation of log-likelihood. We showed in section 7.1 that it outperforms other Transformer-based models on multiple real-world datasets and also beats or ties them on multiple synthetic datasets.

We also showed how to integrate this architecture with NDTT, a neural-symbolic framework that automatically derives neural models from logic programs. Our attention-based modification of NDTT has shown competitive performance, despite having a simpler and shallower architecture. Our code and datasets are available at `https://github.com/yangalan123/anhp-andtt`.

---

[3]While NDTT still uses a fixed-dimensional history—① in section 1—the dimensionality is often very high, as the NDTT's state consists of embeddings of many individual facts. Moreover, each fact's NDTT embedding is computed by rule-specific LSTMs that see only events that are relevant to that fact, so there is no danger that intervening irrelevant events will displace the relevant ones in the fixed-dimensional LSTM states.

## ACKNOWLEDGMENTS

This work was supported in part by the National Science Foundation under Grant No. 1718846. We thank Bloomberg for a Data Science Ph.D. Fellowship to the second author. We thank Minjie Xu for the suggestion of developing a Transformer version of NDTT. We thank the anonymous ICLR 2022 reviewers for discussion and for pointing out additional related work.

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

# Appendices

## A DISCUSSION OF ARCHITECTURAL DETAILS

**Simplification.** Equation (1) is a simplification of the original Transformer architecture (Vaswani et al., 2017). In the original architecture, $[\![e]\!]^{(\ell)}(t)$ would be obtained as $\text{LayerNorm}(\mathbf{x} + \text{FFN}^{(\ell)}(\mathbf{x}))$, where $\mathbf{x}$ is the LayerNorm transformation (Ba et al., 2016) of the right-hand side of equation (1), and the nonlinear transform $\text{FFN}^{(\ell)}$ is computed by a learned two-layer feed-forward network.

In our preliminary experiments, we found that the LayerNorm and FFN steps did not help, so for simplicity and speed, we omitted them from equation (1) and from the remaining experiments. However, it is possible that they might help on other domains or with larger training datasets, so our code supports them via command-line arguments.

**Graceful degradation.** Another change to equation (1) (and equation (8)) is that when normalizing the attention weights, we included an additional summand of 1 in the denominator.[4] We do this so that when the history $\mathcal{H}(e@t)$ is "rather irrelevant" to $e@t$, the architecture behaves roughly as if $\mathcal{H}(e@t)$ were the empty set. In equation (1), this means that $[\![e]\!]^{(\ell)}(t)$ will then be close to $[\![e]\!]^{(\ell-1)}(t)$. Similarly, equation (7) will not be much influenced by rule $r$ if rule $r$ finds only events $\mathcal{H}_r(e@t)$ that it considers to be "rather irrelevant" to $e@t$.

A "rather irrelevant" history is one for which the unnormalized attention weights are small *in toto*, so that the denominator is dominated by the 1 summand. This may occur, for example, if events in the distant past tend to have small attention weights, and the history consist only of old events (and not too many of them). When the history is rather irrelevant, the argument to $\tanh$ in equation (1) and the summand $\boxed{e}_r^{(\ell)}(t)$ in equation (7) are close to $\mathbf{0}$; when $\mathcal{H}(e@t) = \emptyset$, they are exactly $\mathbf{0}$.

**Direct access to time embeddings.** Another difference from Vaswani et al. (2017)—perhaps not an important one—is that in equation (3), we chose to concatenate $[\![t]\!]$ to the rest of the embedding rather than add it (cf. Kitaev & Klein, 2018; He et al., 2020). Furthermore we did so at every layer and not just layer 0. The intuition here is that the extraction of good key and query vectors at each layer may benefit from "direct access" to $[\![t]\!]$. For example, this should make it easy to learn keys and queries such that the attention weight is highest when $s \approx t - \Delta$ (since for every $\Delta \in \mathbb{R}$, there exists a sparse linear operator that transforms $[\![t]\!] \mapsto [\![t - \Delta]\!]$).

**Range of wavelengths for time embeddings.** Our time embedding $[\![t]\!]$ in (4) uses dimensions that are sinusoidal in $t$, with wavelengths forming a geometric progression from $2\pi m$ to $2\pi(5M)$. Setting $m = 1, M = 2000$ would recover the standard scheme of Vaswani et al. (2017) (previously used in continuous time by Zuo et al. (2020)).

We instead set $m$ and $M$ from data so that we are robust to datasets of different time scales. Part of the intuition behind using sinusoidal embeddings is that nearby times can be distinguished by different values in their short-wavelength dimensions, whereas the long-wavelength dimensions make it easy to inspect and compare faraway times, since those dimensions are nearly linear on $t \in [0, M]$. We therefore take $m$ to be the shortest gap between any two events in the same history,

$$m = \min_{e@t} \min_{f@s, f'@s' \in \mathcal{H}(e@t)} |s - s'|, \tag{11}$$

as computed over training data, and take $M$ greater than all $T$ in training data (where each observed sequence is observed over an interval $[0, T]$).

If we were modeling sequences of words as in Brown et al. (2020), our procedure would indeed recover the values $m = 1$ and $M = 2000$ that they used to model text documents. Multiplying or dividing all $t$ values in the dataset by 1000 (e.g., switching between second and millisecond units) would have no effect on the time embeddings, as it would scale $m$ and $M$ in the same way.

**Coarse event embeddings for speed.** As noted at the end of section 3, the intensity model equation (5) involves a full embedding of each $e@t$. This may be expressive, but it is also expensive. The attention

---

[4]Including the summand of 1 is equivalent to saying that $e@t$ attends not only to relevant events $\mathcal{H}(e@t)$ but also to a dummy object whose key and value are fixed at $\mathbf{0}$. The dummy object gets an unnormalized attention weight of 1, drawing attention away from $\mathcal{H}(e@t)$.

weight vectors $\alpha^{(1)}, \ldots, \alpha^{(L)}$ used to compute this embedding must be computed from scratch for each $e$ and $t$. Why is this necessary?

Like other neural sequence models—both RNN-based and Transformer-based—we derive the probability that the next sequence element is $e$ from an inner product of the form $\mathbf{w}_e^\top [1; [\![\mathcal{H}(e@t)]\!]]$, where in our equation (5), the role of the history embedding $[\![\mathcal{H}(e@t)]\!]$ is played by $[\![e]\!]^L(t)$. However, for many previous models, the history embedding does not depend on $e$, so it can be computed once at each time $t$ and reused across all $e$.

- In neural language models, typically *all* previous events are taken to be relevant. $[\![\mathcal{H}(e@t)]\!]$ can then be defined as an RNN encoding of the sequence all past events (Mikolov et al., 2010), or alternatively a Transformer embedding of the single most recent past event (which depends on the entire sequence of past events). This does not depend on $e$.

- When modeling irregularly spaced events, $t$ is not necessarily an integer, and the past events in $\mathcal{H}(e@t)$ do not necessarily take place at $1, 2, \ldots, t-1$. Thus, the encoding $[\![\mathcal{H}(e@t)]\!]$ must somehow be improved to also consider the elapsed time $t - t_i$ since the most recent past event (Du et al., 2016; Mei & Eisner, 2017; Zuo et al., 2020; Zhang et al., 2020a). So now $[\![\mathcal{H}(e@t)]\!]$ must look at $t$, but it is still independent of $e$.

- In contrast, in sections 4–5, we will allow the more general case where $[\![\mathcal{H}(e@t)]\!]$ varies with $e$ as well, since NDTT rules determine which past events should be attended to by $e$. The original NDTT paper essentially defined $[\![\mathcal{H}(e@t)]\!]$ as the state at time $t$ of an $e$-specific continuous-time LSTM, which is updated by just the events that are relevant to $e$. In our attention-based approach, we instead define it to be $[\![e]\!]^L(t)$, yielding equation (5).

To reduce this computational cost, we can associate each event type $e$ with a **coarse event type** $\bar{e}$ that is guaranteed to have the same set of relevant past events, and replace $[\![e]\!]^L(t)$ with $[\![\bar{e}]\!]^L(t)$ in equation (5). (However, equation (5) still uses the fine-grained $\mathbf{w}_e$.) Now to compute $\lambda_e(t)$, we only have to embed $\bar{e}@t$, which is a speedup if many of the possible event types $e$ are associated with the same $\bar{e}$. In the case where we do not use selective attention, we can get away with using only a single coarse event type for the whole model—saving a runtime factor of $|\mathcal{E}|$ as in the cheaper approach. Note that the history $\mathcal{H}$ still uses fine-grained embeddings, so if $e@t$ actually occurs, we must then compute $[\![e]\!]^0(t), \ldots, [\![e]\!]^L(t)$.

**Concatenation vs. summation.** Equation (7) uses summation to combine the outputs (8) of different attention heads $r$. Vaswani et al. (2017) instead combined such outputs by projecting their concatenation, but that becomes trickier in our setting: different events $e$ would need to concatenate different numbers of attention heads $r$ (for just the rules $r$ that can take the form $e \leftarrow \cdots$), resulting in projection matrices of different dimensionalities. Especially when NDTT rules can contain variables (section 5 below), it is not immediately obvious how one should construct these matrices or share parameters among them. These presentational problems vanish with our simpler summation approach.

Our approach loses no expressive power: projecting a concatenation of $\boxed{e}_r^{(\ell)}(t)$ values, as Vaswani et al. would suggest, is equivalent to summing up an $r$-specific projection of $\boxed{e}_r^{(\ell)}(t)$ for each $r$, as we do, where our projection of $\boxed{e}_r^{(\ell)}(t)$ has implicitly been incorporated into the projection (3a) that produces $\mathbf{v}_r^{(\ell)}$. That is, if we can learn $\mathbf{V}_r^{(\ell)} = \mathbf{V}$ in equation (3a), then we can also learn $\mathbf{V}_r^{(\ell)} = \mathbf{PV}$, where $\mathbf{P}$ is the desired projection matrix for rule $r$. To make our method fully equivalent to Vaswani et al.'s, we would have to explicitly parameterize $\mathbf{V}_r^{(\ell)}$ as a matrix product of the form $\mathbf{PV}$, forcing it to be low-rank.

## B  NDTT Example With Variables

The company message forum program in section 5 had only 2 users and 1 forum. However, if the company employs many persons P and has a forum for each team T, NDTT rules can use capitalized **variables** to define the whole system concisely, using only $O(1)$ rules and $O(1)$ parameters. Here the possible facts and events have structured names like `message(eve,sales,joke)`, which denotes an event in which employee `eve` posts a `joke` to the `sales` team's forum.

```
7  forum(T) ← create(T).          9   message(P,T,C) :- empl(P), forum(T), content(C).
8  forum(T) ← message(P,T,C).     10  !forum(T) ← destroy(T).
```

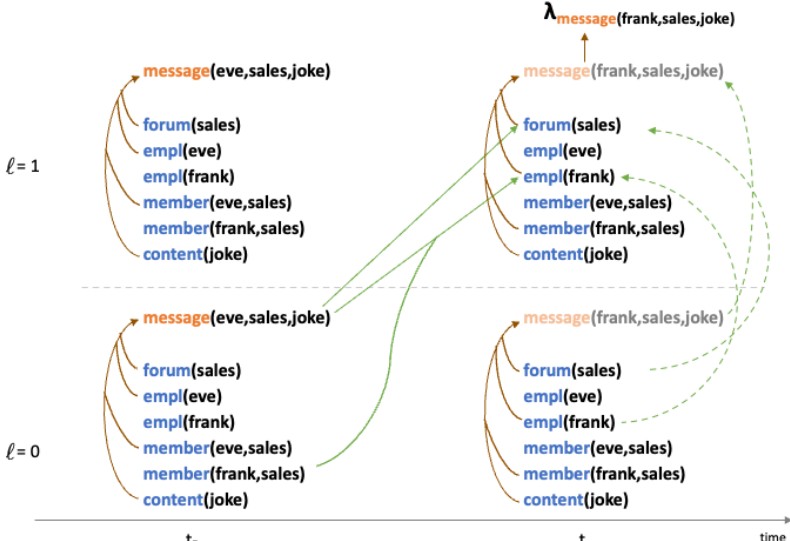

Figure 4: The solid green arrows correspond to instantiations of the attentional $\leftleftarrows$ rules 8 and 13. They can capture the real-world property that thanks to Eve's joke at time $t_5$, the sales forum still feels more humorous at time $t$ and Frank, another member of that forum, is still in a good mood. This raises the probability $\lambda_e(t)\mathrm{d}t$ that Frank posts his own joke at time $t$, where $e = \underline{\texttt{message}}(\texttt{frank},\texttt{sales},\texttt{joke})$. More formally, $\lambda_e(t)$ is determined by the layer-$L$ embedding of Frank's possible message $e$@$t$. In general, the layer-$\ell$ embedding of this message reflects the layer-$(\ell-1)$ embeddings of both Frank and the forum at time $t$, as well as the fact that the possible message is a joke. If the message is actually sent (i.e., the possible event actually happens), its layer-$\ell$ embedding would in turn affect the layer-$(\ell+1)$ embeddings of the forum and its readers at times $> t$. An arrow with multiple inputs means that the input embeddings are concatenated before being transformed into a contribution to the output embedding. Other visual conventions are as in Figure 1. Not all facts, events, or arrows are shown in this drawing.

This generalizes the previous program, allowing multiple forums and saying that any employee (not just Eve and Frank) can post any type of message to any forum, affecting the embedding of that forum. We could modify rule 9 by adding an additional condition `member(P,T)`, so that employees can only post to forums of which they are members. Membership could be established and tracked by

```
11  join(P,T)  :- empl(P), forum(T).
12  member(P,T) ← join(P,T).
13  empl(P) ← message(P2,T,C), member(P,T).
```

Rules 8 and 13 ensure that a message to a forum affects the subsequent embedding of that forum and also the subsequent embeddings of all employees who were members of that forum when the message was sent. This may affect which employees join which forums in future, and what they post to the forums, as drawn in Figure 4 in the appendices. For further examples, see the full presentation of NDTT in Mei et al. (2020a).

How are variables treated in the computation of embeddings? In equations (7)–(8), $r$ refers to a rule with variables. However, $e$ refers to a specific fact, without variables. An **instantiation** of $r$ is a copy of $r$ in which each variable has been consistently replaced by an actual value. In our modified version of equation (8), the summations range over all values of $(f, g_1, \ldots, g_n)$@$s$ such that $e \leftleftarrows f$, $g_1$, $\ldots$, $g_n$ is an instantiation of $r$ that added $e$ at time $s$ (i.e., an instantiation of $r$ such that $f$ occurred at time $s$ and $g_1, \ldots, g_n$ were all true at time $s$). Thus, the attentional competition may consider $(f, g_1, \ldots, g_n)$@$s$ values that are derived from many different instantiations of $r$. Their attentional weights $\alpha_r^{(\ell)}$ are all obtained using the shared parameters associated with rule $r$.[5] The summation in

---

[5] Mei et al. (2020a, Appendix B) provide a notation to optionally reduce the amount of parameter sharing. A rule may specify, for example, that each value of variable `T` should get its own parameters. In this case, we regard the rule as an abbreviation for several rules, one for each value of `T`. Each of these rules corresponds to a different $r$ in equation (7), and so corresponds to a separate attention head (8) that sets up its own attentional

equation (7) ranges only over rules $r$ with at least one instantiation that adds $e@t$, so it skips rules that are irrelevant to $e$.

## C  PARAMETER DIMENSIONALITY SPECIFICATION FOR A-NDTT

In this section we discuss the dimensionality of the fact embeddings $[\![h]\!](t)$ in section 5.

As in the original NDTT paper Mei et al. (2020a), the **type** of a fact in the database is given by its functor (`forum`, `member`, `create`, etc.). All facts of the same type have embedding vectors of the same dimensionality, and these dimensionalities are declared by the NDTT program.

This is enough to determine the dimensions of the parameter matrices associated with the deduction rules (Mei et al., 2020a). How about the add rules, however? The form of equation (9) implies that the value vectors $\mathbf{v}_r^{(\ell)}$ for add rule $r$ have the same dimensionality as the embedding of the head of $r$. The key and query vectors for rule $r$ (as used in equation (2)) can share this dimensionality by default, although we are free to override this and specify a different dimensionality for them. The foregoing choices determine the dimensions of the parameter matrices $\mathbf{V}_r^{(\ell)}, \mathbf{K}_r^{(\ell)}, \mathbf{Q}_r^{(\ell)}$ associated with rule $r$.

## D  LIKELIHOOD COMPUTATION DETAILS

In this section we discuss the log-likelihood formulas in section 3.

Derivations of the log-likelihood formula (6) can be found in previous work (e.g., Hawkes, 1971; Liniger, 2009; Mei & Eisner, 2017). Derivations of this formula appear in previous work (e.g., Hawkes, 1971; Liniger, 2009; Mei & Eisner, 2017). Intuitively, when training to increase the log-likelihood (6), each $\log \lambda_{e_i}(t_i)$ is increased to explain why the observed event $e_i$ happened at time $t_i$, while $\int_{t=0}^{T} \sum_{e=1}^{E} \lambda_e(t)\mathrm{d}t$ is decreased to explain why no event of any possible type $e \in \{1, \ldots, E\}$ ever happened at other times. Note that there is no $\log$ under the integral in equation (6). Why? The probability that there was not an event of any type in the infinitesimally wide interval $[t, t + \mathrm{d}t)$ is $1 - \lambda(t)\mathrm{d}t$, whose $\log$ is $-\lambda(t)\mathrm{d}t$.

The integral term in equation (6) is computed using the Monte Carlo approximation given by Mei & Eisner (2017, Algorithm 1), which samples times $t$. This yields an unbiased stochastic gradient. For the number of Monte Carlo samples, we follow the practice of Mei & Eisner (2017): namely, at training time, we match the number of samples to the number of observed events at training time, a reasonable and fast choice, but to estimate log-likelihood when tuning hyperparameters or reporting final results, we take 10 times as many samples. The small remaining variance in this procedure is shown in our error bars, as explained in footnote 6.

At each sampled time $t$, the Monte Carlo method still requires a summation over all events to obtain $\lambda(t)$. This summation can be expensive when there are many event types. This is not a serious problem for our standalone A-NHP implementation since it can leverage GPU parallelism. But for the general A-NDTT implementation, it is hard to parallelize the $\lambda_k(t)$ computation over $k$ and $t$. In that case, we use the downsampling trick detailed in Appendix D of Mei et al. (2020a).

An alternative would be to replace maximum-likelihood estimation with noise-contrastive estimation, which is quite effective at training NHP and NDTT models (Mei et al., 2020b).

## E  HOW TO PREDICT EVENTS

It is possible to sample event sequences exactly from an A-NHP or A-NDTT model, using the **thinning algorithm** that is traditionally used for autoregressive point processes (Lewis & Shedler, 1979; Liniger, 2009). In general, to apply the thinning algorithm to sample the next event at time $\geq t_0$, it is necessary to have an upper bound on $\{\lambda_e(t) : t \in [t_0, \infty)\}$ for each event type $t$. An explicit construction for the NHP (or NDTT) model was given by Mei & Eisner (2017, Appendix B.3). For A-NHP and A-NDTT, observe that $\lambda_e(t)$ is a continuous real-valued function of $[\![t]\!]$ (the particular function depends on $e$ and the history of events at times $< t_0$). Since $[\![t]\!]$ falls in the compact set $[-1, 1]^d$ (thanks to the sinusoidal embedding (4)), it follows that $\lambda_e(t)$ is indeed bounded. Actual

---

competition using its own parameters. One use of this mechanism would be to allocate multiple attention heads to a single rule.

| DATASET | $K$ | # OF EVENT TOKENS | | | SEQUENCE LENGTH | | |
|---|---|---|---|---|---|---|---|
| | | TRAIN | DEV | TEST | MIN | MEAN | MAX |
| SYNTHETIC | 10 | 59904 | 7425 | 7505 | 49 | 75 | 99 |
| MIMIC-II | 75 | 1930 | 252 | 237 | 2 | 4 | 33 |
| STACKOVERFLOW | 22 | 345116 | 38065 | 97233 | 41 | 72 | 736 |
| ROBOCUP | 528 | 2195 | 817 | 780 | 780 | 948 | 1336 |

Table 1: Statistics of each dataset.

numerical bounds can be computed using interval arithmetic. That is, we can apply our continuous function not to a particular value of $[\![t]\!]$ but to all of $[-1, 1]^d$, where for any elementary continuous function $f : \mathbb{R} \to \mathbb{R}$, we have defined $f([x_{\text{lo}}, x_{\text{hi}}])$ to return some bounded interval that contains $f(x)$ for all $x \in [x_{\text{lo}}, x_{\text{hi}}]$. The result will be a bounded interval that contains $\lambda_e(t)$ for all $t \in [t_0, \infty)$.

Section 7 includes a task-based evaluation where we try to predict the *time* and *type* of just the next event. More precisely, for each event in each held-out sequence, we attempt to predict its time given only the preceding events, as well as its type given both its true time and the preceding events.

We evaluate the time prediction with average $L_2$ loss (yielding a root-mean-squared error, or **RMSE**) and evaluate the argument prediction with average 0-1 loss (yielding an **error rate**).

Following Mei & Eisner (2017), we use the minimum Bayes risk (MBR) principle to predict the time and type with lowest expected loss. For completeness, we repeat the general recipe in this section.

For the $i^{\text{th}}$ event, its time $t_i$ has density $p_i(t) = \lambda(t) \exp(- \int_{t_{i-1}}^{t} \lambda(t')\mathrm{d}t')$. We choose $\int_{t_{i-1}}^{\infty} t p_i(t)\mathrm{d}t$ as the time prediction because it has the lowest expected $L_2$ loss. The integral can be estimated using i.i.d. samples of $t_i$ drawn from $p_i(t)$ by the thinning algorithm.

Given the next event time $t_i$, we choose the most probable type $\arg\max_e \lambda_e(t_i)$ as the type prediction because it minimizes expected 0-1 loss. In some circumstances, one might also like to predict the most probable type out of a *restricted* set $\mathcal{E}' \subsetneq \{1, \ldots, E\}$. This allows one to answer questions like "If we know that some event of the form message(eve,T) happened at time $t_i$, then what was the forum T, given all past events?" The answer will simply be $\arg\max_{e \in \mathcal{E}'} \lambda_e(t_i)$.

## F   EXPERIMENTAL DETAILS

### F.1   DATASET DETAILS

Table 1 shows statistics about each dataset that we use in this paper.

#### F.1.1   PILOT EXPERIMENTS ON SIMULATED DATA

In this experiment, we draw data from randomly initialized NHP, A-NHP, SAHP, and THP. For all of them, we take the number of event types to be $E = 10$. For NHP, the dimensions of event embeddings and hidden states are all 32; for A-NHP, the number of layers ($L$ in our paper) is 2, and the dimensions of time embeddings and event embeddings are 32; for SAHP, the number of layers is 4, and the dimension of hidden states is 32; for THP, the number of layer is 7, and the dimension of hidden states is 32.

For each model, we draw 800, 100, and 100 sequences for training, validation and testing, respectively. For each sequence, the sequence length $I$ is drawn from Uniform(49, 99). We take the maximum observation time $T$ to be $t_I + 1$, one time step after the final event.

We fit all 4 models on each of these 4 synthetic datasets. The results are graphed in Figure 5 and show that NHP, SAHP, and A-NHP have very close performance on all 4 datasets (outperforming THP, especially at predicting timing, except perhaps on the THP dataset itself). Notably, THP fits the time intervals poorly when it is misspecified, perhaps because its family of intensity functions (section 6) is not a good match for real data: THP requires that the intensity of $e$ between events changes more slowly later in the event sequence, and that if it increases over time, it approaches linear growth rather than an asymptote.

Log-likelihood per event (the training objective) of the whole test event sequence

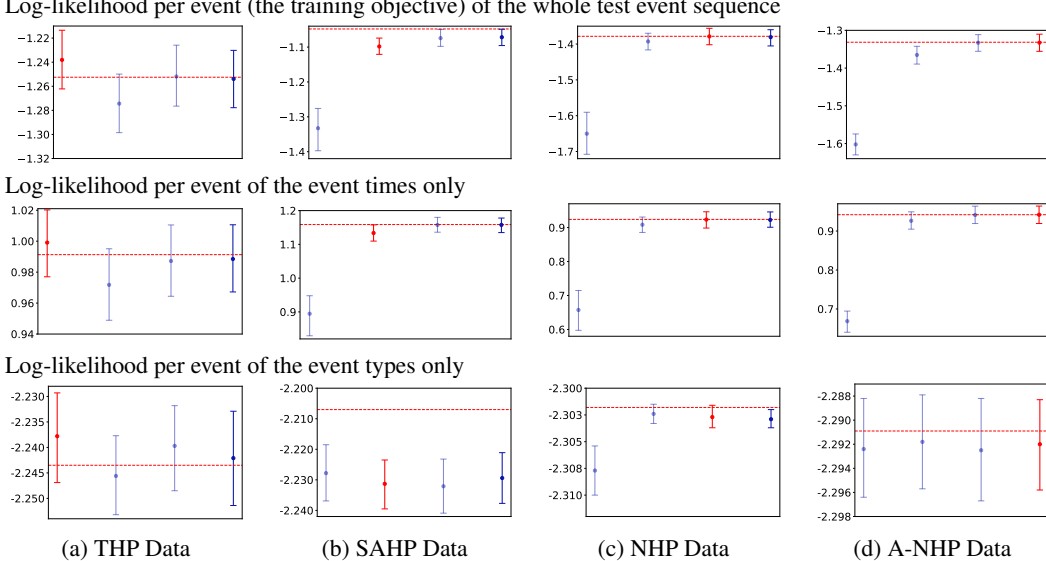

Log-likelihood per event of the event times only

Log-likelihood per event of the event types only

(a) THP Data  (b) SAHP Data  (c) NHP Data  (d) A-NHP Data

Figure 5: Log-likelihood on held-out data (in nats, with 95% bootstrap confidence intervals[6]). Larger values are better. Each column is a different experiment, on a single synthetic dataset generated from a different distribution family (shown at the bottom of the column). Within each column, the red dashed horizontal line represents the log-likelihood of the true distribution that generated the data. Within each column, we train and test 4 models: THP, SAHP, NHP, and A-NHP (from left to right). The model from the correct family is shown in red; compare this to our A-NHP model (the rightmost model). Other models are shown in lighter ink. Note that log-likelihood for continuous variables can be positive (as in the second row), since it uses the log of a probability density that may be $> 1$.

#### F.1.2 OTHER DATA DETAILS

For MIMIC-II and StackOverflow, we used the version processed by Du et al. (2016); more details (e.g., about processing) can be found in their paper.

For RoboCup, we used the version processed by Chen & Mooney (2008); please refer to their paper for more details (e.g., data description, processing method, etc)

#### F.2 IMPLEMENTATION DETAILS

For NHP, our implementation is based on the public Github repositories at `https://github.com/HMEIatJHU/neurawkes` (Mei & Eisner (2017), with MIT License) and `https://github.com/HMEIatJHU/neural-hawkes-particle-smoothing` (Mei et al. (2019), with BSD 3-Clause "New" or "Revised" License). We made a considerable amount of modifications to their code (e.g., model, thinning algorithm), in order to migrate it to PyTorch 1.7. We built the standalone GPU implementation of A-NHP upon our NHP code.

For NDTT, we used the public Github repository at `https://github.com/HMEIatJHU/neural-datalog-through-time` (Mei et al. (2020a), with MIT License). We built A-NDTT upon NDTT.

For THP, we used the public Github repository at `https://github.com/SimiaoZuo/Transformer-Hawkes-Process` (Zuo et al. (2020), no license specified).

For SAHP, we used the public Github repository at `https://github.com/QiangAIResearcher/sahp_repo` (Zhang et al. (2020a), no license specified).

#### F.3 TRAINING DETAILS

For each model in section 7, we had to specify various dimensionalities. For simplicity, we used a single hyperparameter $D$ and took all vectors to be in $\mathbb{R}^D$. This includes the state vectors of NHP, the fact embeddings of NDTT and A-NDTT, and the query, key, and value vectors for the models

| DATASET | TRAINING TIME (MILLISECONDS) / SEQUENCE | |
| --- | --- | --- |
| | NHP | A-NHP |
| SYNTHETIC | 208.7 | 56.3 |
| MIMIC-II | 2.9 | 32.6 |
| STACKOVERFLOW | 156.6 | 65.7 |

Table 2: Training time of NHP and A-NHP for experiments in section 7.1.

with attention mechanisms (THP, SAHP, A-NHP, and A-NDTT). For the models with attention mechanisms, we also had to choose the number of layers $L$.

We tuned these hyperparameters for each combination of model, dataset, and training size (e.g., each bar in Figures 2, 3a and 5), always choosing the combination of $D$ and $L$ that achieved the best performance on the dev set. Our search spaces were $D \in \{4, 8, 16, 32, 64, 128\}$ and $L \in \{1, 2, 3, 4, 5, 6\}$. In practice, the optimal $D$ for a model was usually 32 or 64; the optimal $L$ was usually 1, 2, or 3.

To train the parameters for a given model, we used the Adam algorithm (Kingma & Ba, 2015) with its default settings. We performed early stopping based on log-likelihood on the held-out dev set.

For the experiments in section 7.1, we used the standalone PyTorch implementations for NHP and A-NHP, which are GPU-friendly. We trained each model on an NVIDIA K80 GPU. Table 2 shows their training time per sequence on each dataset.

For section 7.2, we run our NDTT and A-NDTT models only on CPUs. This follows Mei et al. (2020a), who did not find an efficient method to leverage GPU parallelism for training NDTT models. The machines we used for NDTT and A-NDTT are 6-core Haswell architectures. On RoboCup, the training time of NDTT and A-NDTT was 62 and 149 seconds per sequence, respectively. See Appendix F.4 for future work on improving the latter time by exploiting GPU parallelism.

For the NHP and A-NHP models in section 7.2, we ran the specialized code for these models on CPU as well, rather than on GPU as in section 7.1, since the RoboCup sequences were too long to fit in the memory of our K80 GPU. The training time was 66 and 95 seconds per sequence for NHP and A-NHP, respectively.

### F.4 TRAINING PARALLELISM

We point out that in the future, GPU parallelism could be exploited through the following procedure, given a GPU with enough memory to handle long training sequences. (The layers can be partitioned across multiple GPUs if needed.)

For each training minibatch, the first step is to play each event sequence $e_1@t_1, e_2@t_2, \ldots, e_I@t_I$ forward to determine the contents of the database on each interval $(0, t_1], (t_1, t_2], \ldots, (t_{i-1}, t_I], (t_I, T]$. This step runs on CPU, and computes only the boolean facts ("Datalog through time") without their embeddings ("neural Datalog through time").

Let $\mathcal{F}$ be the set of facts that ever appeared in the database during this minibatch and let $\mathcal{R}$ be the set of rules that were ever used to deduce or add them (section 5). Furthermore, let $\mathcal{T}$ be the set of times consisting of $\{t_1, \ldots, t_I\}$ together with the times $t$ that are sampled for the Monte Carlo integral (Appendix D).

A computation graph of size $O(|\mathcal{R}| \cdot I)$ can now be constructed, as illustrated in Figure 1b, to compute the embeddings $[\![h]\!](t)$ of all facts $h \in \mathcal{F}$ at all times $t \in \mathcal{T}$. The layer-$\ell$ embeddings at time $t \in \mathcal{T}$ depend on the layer-$(\ell - 1)$ embeddings at times $t_i \leq t$, according to the add rules in $\mathcal{R}$. The layer-$\ell$ embedding of a fact that is deduced at time $t$ also depends on the layer-$\ell$ embeddings at time $t$ of the facts that it is deduced from, according to the deduction rules in $\mathcal{R}$; this further increases the depth of the computation graph.

For a given fact $h \in \mathcal{F}$, $[\![h]\!]^{(\ell)}(t)$ can be computed in parallel for *all event sequences* and *all times* $t \in \mathcal{T}$ (even times $t$ when $h$ is not true, although those embeddings will not be used). Multiple facts that are governed by the same NDTT rule $r \in \mathcal{R}$ can also be handled in parallel, since they use

the same $r$-specific parameters. Thus, a GPU can be effective for this phase. The computation of $\boxed{h}_r^{(\ell)}(t)$ in equation (9) must take care to limit its attention to just those earlier times when an event occurred that added $h$ via rule $r$, and the computation of $[h]^{(\ell)}(t)$ in equation (9) must take care to consider only rules $r$ that in fact deduce $h$ at time $t$ because their conditions are true at time $t$. This masks unwanted parts of the computation, rendering parts of the GPU idle. GPU parallelism will still be worthwhile if a substantial fraction of the computation remains unmasked—which is true for relatively homogenous settings where most facts in $\mathcal{F}$ hold true for a large portion of the observed interval $[0, T)$, even if their embeddings fluctuate.

### F.5 MORE RESULTS

The performance of A-NDTT and NDTT is not always comparable for specific action types, as shown in Figure 6. In terms of data fitting (left figure), A-NDTT is significantly better at the `kickoff` events while NDTT is better at the others. For time prediction (middle figure), A-NDTT is significantly better at the `goal`, `kickoff`, and `pass` events, but the differences for the other action types are not significant. For action participant prediction (right figure), A-NDTT is significantly better at the `kickoff` events while there is no difference for the others; both do perfectly well at the `goal` and `kick` events such that their dots overlap at the origin.

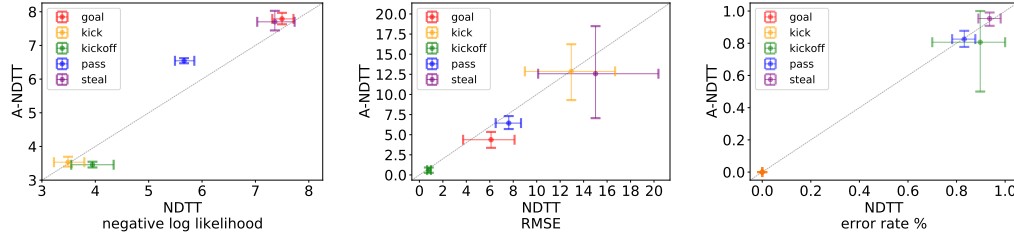

Figure 6: Results of NDTT and A-NDTT in Figure 3a broken down by action types, with horizontal and vertical error bars, respectively.

In Figure 7, we show that Figure 6 does not change qualitatively when re-run with different random seeds.

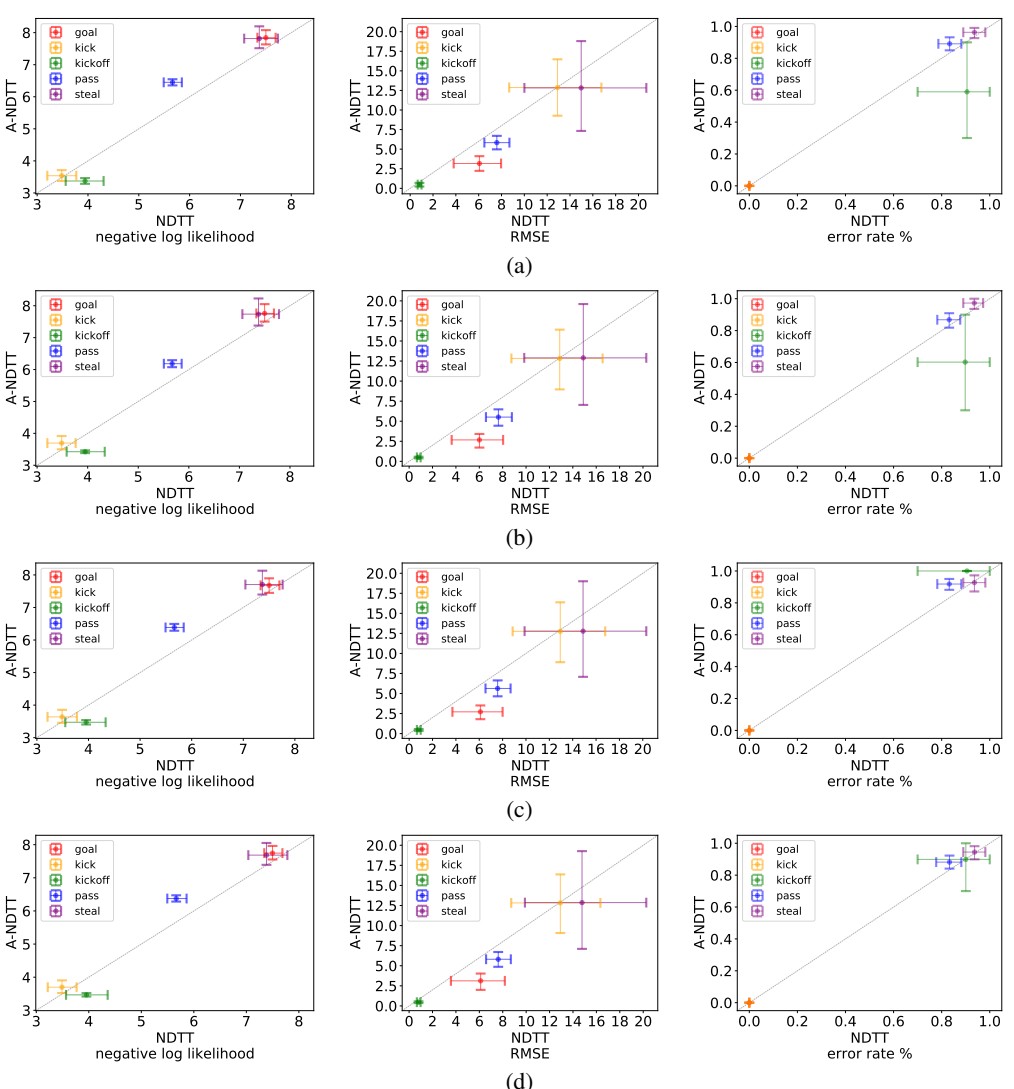

Figure 7: Replications of Figure 6 (one per row) with different random seeds used during training.

