# OpenReview forum: "Transformer Embeddings of Irregularly Spaced Events and Their Participants"
_ICLR.cc/2022/Conference — ICLR 2022 Poster_

### Official Review · Reviewer_9eGx · 2021-11-02

**Correctness:** 4
**Technical Novelty And Significance:** 2
**Empirical Novelty And Significance:** 2
**Recommendation:** 3
**Confidence:** 3

**Main Review:**

Overall, the paper provides some interesting contributions on modeling continuous-time event processes and adding specifications via the NDTT framework.
However, the paper also contains several weaknesses, which makes it not ready for publication in its current format.
The writing needs to be substantially improved before this manuscript is ready for publication. What is the intuition on Eq (1-4)? Why would the embeddings of the time be relevant from the second layer onward?  It is unclear what the main contribution of the paper is. On the one hand, the paper proposes continuous-time event transformers, while on the other hand, the paper proposes a neuro-symbolic framework for embedding rules of which event can draw information from which other events.
The paper assumes the events are discrete-valued, i.e., $e \in \{1,2,\dots E\}$, which limits its scopes for general-purpose modelling.
The paper fails to experimentally compare the proposed continuous-time transformer to existing continuous-time models on standard benchmarks (PhysioNet, etc.) [1,2,3,4]
The paper fails to compare with existing continuous-time attention-based architectures [5].
From the experiments' side, there seems only little to no improvement over standard NDTT.
No code was provided.

[1] Rubanova et al. Latent Ordinary Differential Equations for Irregularly-Sampled Time Series. NeurIPS 2019
[2] Singh et al. Sequential neural processes. NeurIPS 2019.
[3] Li et al. Scalable gradients for stochastic differential equations. AISTATS 2020.
[4] Norcliffe et al. Neural ODE Processes. ICLR 2021
[5] Shukla and Marlin. Multi-Time Attention Networks for Irregularly Sampled Time Series. ICLR 2021

**Summary Of The Paper:**

The paper proposes a continuous-time transformer architecture for event sequence tasks using the Neural Datalog Through Time (NDTT) framework for incorporating logic specifications into the model.
The paper then experimentally compares the proposed model with point-process models and standard NDTT.

**Summary Of The Review:**

I recommend the authors to improve the writing and clarify whether their approach aims to contribute to general-purpose continuous-time modeling or on incorporating continuous-time logic specification via the NDTT framework. In the current form, neither of these two objectives is fulfilled adequately (see comments above).

---

> ### Author Response · Authors · 2021-11-23
> **about architectural design**
>
> Thanks for raising your concerns about our architectural design and pointing out references. Please find our point-by-point response below.  We hope that this addresses your concerns.
>
> > What is the intuition on Eq (1-4)?
>
> Eq (1-4) is just a simplified version of the standard Transformer, slightly modified to use continuous time.  To better understand the Transformer, we recommend the original paper (Vaswani et al., 2017), as well as Jay Alammar’s post The Illustrated Transformer.
>
> > Why would the embeddings of the time be relevant from the second layer onward?
>
> For the same reason that skip connections can help in a feedforward net.  It is reasonable to imagine that layer 3 cares about the order and recency of events at layer 2.  While layer 2 could certainly learn to preserve timing information from earlier layers, making that information directly accessible increases the chance that it will be used.
>
> > It is unclear what the main contribution of the paper is. On the one hand, the paper proposes continuous-time event transformers, while on the other hand, the paper proposes a neuro-symbolic framework for embedding rules of which event can draw information from which other events.
>
> Please see [Main Contributions] above for our detailed response.
>
> > The paper assumes the events are discrete-valued, i.e., e∈1,2,…E, which limits its scopes for general-purpose modelling.
>
> NHP uses an unstructured discrete event vocabulary {1,2,…E}.  However, NDTT permits a large vocabulary of events in which similar events have related structured names.  Many complex practical domains can be formulated in this way.  NDTT makes it possible to exploit the structure in such domains through shared parameters and shared independence assumptions.
>
> Furthermore, the discrete events may carry metadata ("marks") that include continuous values.  For example, an event where a patient gets a blood test may include the results of the blood test.  This extension is explained in many previous papers on point processes (including the NDTT paper), and is orthogonal to our contribution here.  It makes our approach more broadly applicable.
>
> > No code was provided.
>
> The reproducibility statement says: "we are committed to releasing our code when the paper is published".

---

> > ### Comment · Reviewer_9eGx · 2021-11-29
> > **Not convincing**
> >
> > I have read the author's response, but I am not convinced by it.
> >
> > First of all, the mentioned related works propose models for continuous-time event sequences, i.e., as in the submitted paper. Claiming that these works are not relevant seriously undermines the research done in these papers.
> >
> > Moreover, the author's response to my questions where inadequately answered.
> > I didn't ask where the Eq (1-4) originate, but what is the general intuition of this adaptation of the Transformer. Why would this particular modification work?
> > Regarding the embeddings in the following layers, as the Transformer has skip connections, I don't think the authors make a valid point here. An ablation study would have clarified this.
> >
> > Finally, the statement about the code release undermines thorough peer-review, by keeping the code closed as *bargaining* material.
> >
> >
> > I will keep my recommendation for a rejection.

---

> > > ### Author Response · Authors · 2021-11-30
> > > **about suggested works, architectural design and code release policy**
> > >
> > > 1) The work you mentioned is sound, of course!   Our response merely pointed out that it deals with quite different kinds of data, for which different kinds of models are appropriate.  That was not a criticism of the research in those papers.
> > >
> > >     We certainly aren’t claiming to be comparable to all continuous-time models (e.g., Gaussian processes, censored SDEs, etc.).  Rather, we are extending the long line of work on temporal point processes.  Surely these different lines of work can coexist, without having to cite one another continually?  Past papers in these lines of work have not cross-cited in this way, as far as we know, so it seems odd that you think it necessary in this case.
> > >
> > >     Perhaps you are reacting to some line in our paper that appears to claim more territory than we intended?  We will make sure that our claims are clearly limited to autoregressive temporal point processes, i.e., generative models over sequences of (discrete event, continuous time) pairs.
> > >
> > > 2. Regarding (1)-(4), sorry for misunderstanding your question.  We did share our intuition in our response. We don’t feel strongly about concatenating the timing information at every layer; it was just a minor and very inexpensive architectural choice that we made in hopes that it might help, and perhaps not an important one (you’re correct that skip connections already carry the timing info forward).  There is prior work, which we should have cited, that also adds timing information for every layer ([1], equations (4)-(5); [2], equation (12)), so we think that this was in the space of reasonable discretionary architectural choices.  We agree that a careful investigation of how and when to add timing information for both discrete-time and continuous-time transformers would be a useful separate engineering paper.
> > >
> > >     [1] Dehghani, Mostafa, et al. "Universal Transformers." International Conference on Learning Representations. 2018.  https://arxiv.org/pdf/1807.03819.pdf
> > >
> > >     [2] Liu, Xuanqing, et al. "Learning to encode position for transformer with continuous dynamical model." International Conference on Machine Learning. PMLR, 2020. https://arxiv.org/pdf/2003.09229.pdf
> > >
> > > 3. Regarding code release: Your original review only wrote “No code was provided.”  We thought you were complaining that the paper is less valuable because it doesn't come with software.  Hence in response, we pointed out that our submission did promise to release tcode.  Releasing mature software together with the camera-ready version is good citizenship, so we try to follow this standard practice.
> > >
> > >     Are you instead complaining that you would have liked to examine the code as part of your review? If so, sorry about that, and we applaud your thoroughness.  If you tell us what you wanted to ascertain from the code, we can try to help by providing more information or pointing to existing information in the writeup.  Please don’t assume bad motives on our part.  We tried to include full information for replicability in the paper and appendices.  The only reason we didn’t include the code as well is that sanitizing and anonymizing our codebase would have caused us to miss the deadline in this case.
> > >
> > >     (Or are you taking the stand that all submissions should include code on pain of rejection?  That is stronger than ICLR's policy.  ICLR’s call for papers did encourage supplementary code submission to facilitate review, but made it optional.  Reviewers rarely seem to look, anyway ...)
> > >
> > > 4. Might other reviewers or the AC be willing to mediate on these issues?  We will follow their recommendations.

---

> ### Author Response · Authors · 2021-11-23
> **about suggested references**
>
> > The paper fails to experimentally compare the proposed continuous-time transformer to existing continuous-time models on standard benchmarks (PhysioNet, etc.) [1,2,3,4] The paper fails to compare with existing continuous-time attention-based architectures [5].
>
> With apologies, we are having trouble seeing the relevance of these papers.  They seem to us to be dealing with quite different problems.
>
> [1] uses a VAE framework implemented using ODE-RNN to introduce structures for point process data. Irregular sampling of an SDE is a different setting -- that's partial observation of a function that is underlyingly continuous, where the observation times are chosen outside the model.  Our model deals with full observation of an irregular time series, where the event times are chosen within the model.
>
> Another difference from [1] is that our model does not introduce latent structures using VAE.  We allow domain experts to create an explicit structure that influences embeddings and attention, but this is a structure on observed events.
>
> [2] and [4] work on dynamic scene modeling and image generation tasks.  Those are regression tasks in CV and it is not clear why these two works are relevant.
>
> [3] proposes a new memory-efficient optimization method to solve gradients for SDE using higher-order solvers. Our model does not use SDE so it is also not clear to us why this paper is relevant.
>
> [5] works on multi-time series data, where one target value is associated with multiple partially-observed time-series data. Also, they are building a discriminative model, instead of proposing a probabilistic model. They do discretization over time-series to make them be of roughly the same length. Given these differences, we do not think [5] is relevant.

---

### Official Review · Reviewer_2v8P · 2021-11-04

**Correctness:** 3
**Technical Novelty And Significance:** 2
**Empirical Novelty And Significance:** 2
**Recommendation:** 5
**Confidence:** 4

**Main Review:**

**Strengths**
- The proposed model is well-motivated.
- The architecture and possible variations are described very thoroughly.
- Combining the model with logical rules using the NDTT framework (Section 5) allows to incorporate known constraints on the modelled environment, which may lead to better generalization and predictive performance (Section 8).

-------

I have the following main concerns regarding this paper:

1. **Missing discussion of closely related works**

- The idea of using transformer architectures to define neural TPP models has been explored in multiple existing works. While the THP (Zuo et al., ICML 2020) and SAHP (Zhang et al., ICML 2020) models have been mentioned in this paper, several other published works have been missed and not compared with:

    - [Enguehard et al., ML4H 2020](https://arxiv.org/abs/2007.13794)
    - [Chen et al., ICLR 2021](https://arxiv.org/abs/2011.04583)
    - [Zhu et al., AISTATS 2021](https://arxiv.org/abs/2002.07281)
    - [Sharma et al., KDD 2021](https://arxiv.org/abs/2008.11308)

- The definition of selective attention introduced in Section 4 is equivalent to the notion of Granger causality. GC has been extensively studied in the field of marked TPPs and has been combined with transformer-based neural TPP models ([Zhang et al., WWW 2021](https://dl.acm.org/doi/fullHtml/10.1145/3442381.3450135)).


2. **A more thorough empirical evaluation is needed to support the main claims of the paper**

- One of the claimed advantages of the proposed A-NHP model is its **flexibility** (Section 1, point 3; Section 7). However, the experiments only consider THP and SAHP as baselines. These model assume a rather simple parametrization of the inter-event time distribution, which limits their flexibility, as pointed out in Section 7. There exist multiple more flexible architectures, such as [(Omi et al., 2019)](https://arxiv.org/abs/1905.09690), [(Shchur et al., 2020)](https://arxiv.org/abs/1909.12127) or [(Sharma et al., 2021)](https://arxiv.org/abs/2008.11308), that haven't been compared with.

    These models have other advantages compared to the A-NHP model such as closed-form likelihood computation and lower memory footprint, while A-NHP requires multiple forward passes to evaluate the intensity at random points during Monte Carlo approximation of the integral. Therefore, comparing to these flexible and more efficient models would be important to answer if the higher runtime/memory consumption of A-NHP is justified by its improved flexibility.

----

**Minor suggestions**
- Section 5: The contents are rather dense and are difficult to comprehend without first reading Mei et al., 2020a. Adding a figure that demonstrates how the rule embedding $[\\![h]\\!]$ $(t)$ affects the conditional intensity for different event types would be really helpful (similar to Figure 1b).

- Figure 2: The figure is difficult to read. Color-coding or using symbols to denote different models in each plot would make it easier to understand.

- Reducing the amount of new / non-standard notation, if possible, would improve the readability of the paper.

- Another claim regarding the A-NHP model that is not explored in the experiments is its claimed ability to learn long-range interactions better than an RNN-based TPP, like NHP (Section 1, point 2). It would be interesting to evaluate this aspect of the model and highlight these failure modes of RNN-based TPPs. One idea would be to do this using multivariate Hawkes processes with long-range triggering kernels and many marks.


------
## Update after the rebuttal

The rebuttal has addressed some of my concerns, so I have raised my score
- An efficient approximation to the log-likelihood is available, so the increased runtime / memory consumption compared to existing models shouldn't be a problem.
- The clarity of writing in Section 5 has been improved

Still, I have several remaining concerns, which is why I'm leaning towards rejection. Here are my suggestions for improving the paper:
- Comparison with additional baselines, such as ADMN and LogNormMix
- Make the connection to prior work more explicit in Section 4 (selective attention), add citations to other works on transformer-based TPPs, and better frame the contribution of this paper with respect to prior works
- Explain how the upper bound for the thinning algorithm is derived for A-NTPP, since this is a crucial implementation detail

**Summary Of The Paper:**

The main contribution of this work is a new transformer-based temporal point process (TPP) model.
The proposed model defines the conditional intensity at each time $t$ as a function of all past events using the attention mechanism.
This is different form existing neural TPP models that typically embed the event history into a vector.
Additionally, it is shown how the proposed model can be combined with the Neural Datalog Through Time (NDTT) framework [(Mei et al., 2020)](https://arxiv.org/abs/2006.16723), which allows us to specify certain hard constraints on the event occurrences (such as "events of type x can only occur when certain conditions are met").

**Summary Of The Review:**

My main criticism of this paper is the lack of discussion and comparison with existing works, as described in point 1 and 2 above. The proposed modifications to the NDTT framework [(Mei et al., 2020)](https://arxiv.org/abs/2006.16723) don't lead to significant improvements over the original methods.

---

> ### Author Response · Authors · 2021-11-23
> **about suggested references**
>
> Thank you for suggesting references. Some seem more relevant than others, as we discuss below. We also answer your other questions (as well as giving some extra experimental results).
>
> > Chen et al., ICLR 2021
>
> The work deals with a different kind of data.  We can’t run on their data and they can’t run on ours.  In their setting, each event is labeled with a time and a continuous spatial location.  Their methods use normalizing flows and appear to be specialized to continuous location.  In our setting, however, each event is labeled with a time and a discrete structured event type.
>
> > Enguehard et al., ML4H 2020
>
> This work is related and we will add this citation.  However, this work appears to be very similar to THP, which we did compare with. Both models only incorporate time information in temporal embeddings in the input layer, so their attention vector is constant between events.
>
> > Sharma et al., KDD 2021
>
> This paper seems to be an application of temporal point process models (they use a model very similar to THP), rather than a new model in this family.  The main goal of their paper is to identify latent coordinated accounts in social media.  Their method cannot be evaluated on our data.
>
> > Zhu et al., AISTATS 2021
>
> Thanks, good catch.  At the start of their section 3.2, they do mention in passing a model very similar to our A-NHP model (except that their version oddly uses the same matrix to extract attentional queries and keys, which means that attention will prefer similar events).  However, they don't evaluate this model at all.
>
> Our A-NDTT model also continues in a direction quite different from theirs.  We show how to make use of the expert-written rules from Mei et al. (2020) to limit and specialize attention in a domain-specific way (see [Main Contributions] above).
>
> > These models have other advantages compared to the A-NHP model such as closed-form likelihood computation and lower memory footprint
>
> Perhaps so.  Closed-form computation and lower memory footprint are also advantages of linear regression over deep learning.  However, deep learning is occasionally better at fitting the data. :-)  And A-NHP is better than THP, too.  We agree that it’s worth pointing out the strengths and weaknesses of different models.

---

> > ### Comment · Reviewer_2v8P · 2021-11-29
> > **Reply to the authors' response #3**
> >
> > The reason I mention all these works is the phrase `Yet there exists little work on generalizing its success to sequences of discrete
> > events in continuous time` in the abstract of the paper. To my knowledge, there exist at least 6 published papers that generalize the transformer architecture to modeling continuous-time event sequences (4 of which have not been cited in this submission), so this statement in the abstract seems inaccurate to me.
> >
> > Also, I want to mention that the paper by Sharma et al., KDD 2021 introduces a new transformer-based TPP architecture (AMDN). While it encodes the event history into a vector, similar to other transformer-based models, they use a different parametrization of the inter-event time distribution, which might lead to noticeable improvements in the NLL. It should be possible to compute NLL, MSE and accuracy metrics for this model since the code is available: https://github.com/USC-Melady/AMDN-HAGE-KDD21.
> >
> > Finally, the comparison to linear regression seems inappropriate to me: Other transformer-based architectures are also based on deep learning; the proposed A-NHP model requires $S$ times more memory & time due to additional calls to self-attention (so it might be unable to handle long sequences, see reply #1), and doesn't permit analytic likelihood computation or sampling. While the A-NHP model may be more flexible — which undoubtedly is an advantage — these trade-offs compared to other models should be discussed.
> >
> > -----
> > To summarize my replies to parts 1/2/3 of the rebuttal: I'm grateful for your detailed response, but it has not addressed my concerns with the current submission, so I would like to keep my current rating.

---

> > > ### Author Response · Authors · 2021-11-30
> > > **thanks for the explanation**
> > >
> > > Got it: thanks very much for the explanation.  We apologize for the overclaim in the abstract and will remove it, as well as adding discussion of the papers that are relevant to our type of setting (as detailed in our previous reply).
> > >
> > > You are correct that Sharma et al. do use an interesting novel TPP parameterization, so we’ll cite that as an alternative approach, although unfortunately they have not evaluated it in isolation or compared it to other methods.
> > >
> > > A-NHP actually does allow exact sampling (via the thinning algorithm, the same as NHP).  As you say, it does not allow exact log-likelihood computation, although it does have a simple unbiased stochastic estimator of log-likelihood.  Regarding the factor of S, see our separate response above.

---

> > > > ### Comment · Reviewer_2v8P · 2021-12-01
> > > > **Response from the reviewer**
> > > >
> > > > I will summarize all the points here:
> > > >
> > > > 1. Number of MC samples: Thanks for clarifying this, after reading your explanation, I agree that training with $S = 1$ won't incur much of an overhead in terms of time or memory. However, it would be important to use a much larger value of $S$ when comparing the NLL in Section 8, and, potentially, ensure that the variance in the estimate of the NLL due to different samples is reflected by the error bars.
> > > > 2. Selective attention: I see that NDTT is more general than the influence matrix used in previous works. However, the description in Section 4 only refers to constraining influence between different event types, which is equivalent to the enforced sparsity pattern on the influence matrix in previous works. Therefore, presenting selective attention as a novel contribution in Section 4 seems inaccurate.
> > > > 3. Sampling: As far as I understand, neither NHP or A-NHP permit exact sampling via thinning, since it's impossible to obtain an upper bound on the intensity. Even if we compute intensity at several points sampled from $(t_i, \infty)$, there is no guarantee that the intensity isn't higher at some other point that wasn't sampled, so the upper bound can't be computed, hence thinning cannot be done correctly. The only plausible alternative I see is to approximate the compensator via MC & try to invert it with numerical root finding, which is less accurate and less efficient than methods permitting analytic sampling. Please correct me if I misunderstand something here.
> > > >
> > > > ----
> > > > To summarize, I see the value of the proposed model thanks to its high flexibility, and you have addressed some of my concerns with the rebuttal. Still, I have multiple remaining concerns that would require substantially changing the paper, and therefore cannot be addressed in this discussion period:
> > > > - Some contributions, such as selective attention in Section 4 (not A-NDTT in Section 5), are presented as novel, even though they are very similar to the published work.
> > > > - Connections & limitations with respect to other existing methods are not discussed, and some baselines are missing in the experiments.
> > > > - A more intuitive explanation of the contents of Section 5 would improve the readability of the paper, as also pointed out by other reviewers.

---

> > > > ### Author Response · Authors · 2021-12-03
> > > > **thinning algorithm, prior work discussions**
> > > >
> > > > (This is the response for comments posted below at https://openreview.net/forum?id=Rty5g9imm7H&noteId=y1HR5j2oQ_N)
> > > >
> > > > Thanks for continuing the discussion; this is fun and a real advantage of OpenReview.
> > > >
> > > > 1. Indeed, for evaluation on dev and test data we used $S=10$ for this reason (following Mei & Eisner 2017 who did the same).  You make a good point that our error bars on the estimated log-likelihood should be revised to reflect any additional variance due to $S$ being too small.  That will be straightforward to handle during the construction of our bootstrap confidence interval: each time a sequence is selected in a bootstrap replicate, its MC times should be redrawn (either fresh, or with replacement from the set of times that were used to estimate the log-likelihood for that sequence, which is faster because their intensities were already computed). For our present settings, the effect is likely to be imperceptible.
> > > > 2. In section 4, we will make explicit the connection to prior work on sparse influence matrices in temporal models.  Section 4 didn’t actually claim to be novel and we didn’t intend it as one of our contributions; we didn’t even give it a name.  We included it for expository reasons, as a transition from A-NHP (section 3) to A-NDTT (section 5).  The <- notation is introduced in its simplest form in section 4 and is then enriched in section 5.
> > > > 3. Thanks for asking.  An upper bound on the intensity over $(t_i,\infty)$ actually can be computed for both models.  Mei & Eisner (2017, Appendix B.3) give the explicit construction for NHP.  For A-NHP, the conditional intensity of event e@t is a continuous function applied to a subset of the compact domain $[-1,1]^d$ (thanks to the sinusoidal embedding of $t$ into that domain), and is therefore bounded.  Actual numeric bounds can be computed using interval arithmetic (just as for NHP).  Thus, the thinning algorithm is exact (assuming a source of true randomness).
> > > >
> > > > Finally, we already submitted a revision during the response period with a longer and more intuitive version of section 5.  We hope you find it more readable.  The new material is marked in the PDF with an olive sidebar.
> > > >
> > > > We intend to incorporate other feedback from this discussion, if you trust us to do so.

---

> > > > > ### Comment · Reviewer_2v8P · 2021-12-07
> > > > > **Response from the reviewer**
> > > > >
> > > > > Thanks a lot for the clarification. Please see the updated review for more details.

---

> ### Author Response · Authors · 2021-11-23
> **Granger causality test vs. independence assumption**
>
> > The definition of selective attention introduced in Section 4 is equivalent to the notion of Granger causality.
>
> Not really.  Granger causality is a TEST on the data.  Selective attention is more like a conditional independence ASSUMPTION about the data, stated by a domain expert.  To omit f <- e from the model says that neither the probability of f nor the embedding of f is affected by e.
>
> Perhaps you meant that if omitting f <- e from the model harms prediction of f, then we can conclude that e events Granger-causes f events.  But that is not necessarily true.  Omitting f <- e from the model might harm the prediction of f events indirectly, by harming the embedding of f events and thus making it hard to use them to predict future f events.

---

> > ### Comment · Reviewer_2v8P · 2021-11-29
> > **Reply to the authors' response #2**
> >
> > Sorry, I guess my explanation wasn't clear. I agree that the notion of Granger causality, technically, refers to the statistical test that aims to determine whether certain event types / dimensions in a multivariate time series are independent. However, in context of marked TPPs the term "Granger causality matrix" or "influence matrix" $G$ usually refers to a $K \times K$ matrix encoding the conditional independence assumption between event types (here $K$ is the # of event types). For example, in the (Zhang et al., 2021, WWW) paper, setting the entry $G_{f,e} = 0$ prevents events of type $f$ from attending to past events of type $e$, which means that the embedding for mark $f$, and hence the conditional distribution $p^*(t_i | v_i = k)$ are unaffected by past events of type $e$. Such constraint can be explicitly specified based on domain knowledge, as explained in Section 4.2 of their paper. The same principle has been extensively used in earlier works on self-exciting TPPs, where an a priori known graph structure (e.g., structure of the social network) is used to prevent possible interactions between users who are not connected (e.g., https://arxiv.org/abs/1507.02293).
> >
> > To me, the concept of selective attention introduced in Section 4 of your paper seems extremely close to the above-mentioned works, and I believe that this connection should be discussed.

---

> > > ### Author Response · Authors · 2021-11-30
> > > **selective attention**
> > >
> > > Thanks for the detailed explanation! Indeed, selective attention is specifying the sparsity pattern of the influence matrix. Importantly, in a deep TPP, this sparsity pattern also affects the construction of event embeddings, not just event intensities as in traditional TPPs.  You are correct that this is also true for Zhang et al. (2021) and we’ll cite them.
> > >
> > > Thanks also for the reference to Farajtabar et al. (2016) and related papers; we should cite this line of work as relevant background for NDTT.
> > >
> > > We do note that NDTT and A-NDTT are somewhat more general, in the following ways:
> > >
> > > - Their “influence matrix” is facts x facts, not just events x events. (All facts have embeddings but only the facts corresponding to possible events have intensities.)
> > >
> > > - A fact may “influence” contemporaneous facts via :- rules as well as later facts via -> rules, which extends the usual notion of influence.
> > >
> > > - Each rule defines the sparsity pattern (and parametric form) of its own influence matrix, giving rise to “multi-headed selective attention.”
> > >
> > > - These matrices may be quite large, and yet can be estimated in practice, because the rule enforces parameter sharing among similarly named facts.
> > >
> > > Thus, we hope that by “extremely close” you don’t mean to imply that our framework adds nothing interesting.

---

> ### Author Response · Authors · 2021-11-23
> **computation, presentation, and experiments**
>
> > while A-NHP requires multiple forward passes to evaluate the intensity at random points during Monte Carlo approximation of the integral.
>
> In A-NHP, we only need **ONE** forward pass to evaluate the intensities at the random points used to compute the MC approximation of the integral. We do not compute the intensities of such points one by one. Instead, we concatenate the observed points and those random points, feed them to our transformer model and use the masking to specify the attended range of each point, which allows the intensities computation to run in parallel in GPU and thus is highly efficient. This follows the code
>
> > Adding a figure that demonstrates how the rule embedding [[h]](t) affects the conditional intensity for different event types would be really helpful (similar to Figure 1b). Figure 2: The figure is difficult to read. Color-coding or using symbols to denote different models in each plot would make it easier to understand.
>
> Thanks for the advice.  We’ll make a new figure and improve the existing figures in the camera-ready.
>
> > It would be interesting to evaluate this aspect of the model and highlight these failure modes of RNN-based TPPs. One idea would be to do this using multivariate Hawkes processes with long-range triggering kernels and many marks.
>
> Thank you for this suggestion! We actually did synthetic experiments of that style but just didn't include them in the paper.
>
> Specifically, we draw sequences from a "skip-gram" process.  The type of each event depends on the types of previous---but not most recent---events. We found that our Transformer-based A-NHP indeed better captured such synthesized long-range dependencies than NHP (which is LSTM-based): A-NHP achieved significantly higher log-likelihood (-0.86 vs -0.93) and lower RMSE for time prediction (2.64 vs. 7.55); A-NHP has a higher error rate on type prediction (8.62% vs 7.47%).

---

> > ### Comment · Reviewer_2v8P · 2021-11-29
> > **Reply to the authors' response #1**
> >
> > Thank you for your response. I understand that the intensities can be computed simultaneously for all the randomly sampled points. However, the self-attention operation needs to be performed for all randomly sampled points, which leads to the memory & time complexity of $O(N^2S)$ (where $N$ is the # of events in the sequence and $S$ is the number of MC samples). In contrast, other attention-based models, such as THP, SAHP and ADMN have complexity of $O(N^2)$ — self-attention is computed once for each event, not $S$ times per event. These models already struggle to deal with medium-sized sequences (1K+ events), and I suspect that this problem is even more pronounced for the A-NHP model due to the additional factor $S$. This is my concern that is not currently addressed by the experiments or the discussion in the paper.

---

> > > ### Author Response · Authors · 2021-11-30
> > > **the number of MC samples**
> > >
> > > We assume you mean O(N^2 (1+S)) where S is the number of sampled times per observed event (i.e., there are NS sampled times in total).  S is usually taken to be a small constant. Mei & Eisner (2017) took S=1 during training to balance the number of positive and negative samples, reporting that this was “large enough for stable behavior.”  We followed this recommendation in our experiments.  Note that any value of S gives an unbiased stochastic gradient.  Mei, Wan & Eisner (2020) give a careful discussion of MLE and NCE asymptotic and empirical runtime for NHP and NDTT.  For MLE, they show learning curves (accuracy vs. training time) for various values of S (which they call rho) that they say were “among the better ones that we found during hyperparameter search.”  All of these curves have S <= 2 and often have S << 1.  Thus, the additional factor S should not be a problem.
> > >
> > > Does this resolve your concern?  In our submission, we reported significant improvements in accuracy over THP and SAHP; we are happy to add an appendix with a runtime comparison if you ask us to do so.  Note that our submission did not ignore runtime concerns; we provided (and implemented) an orthogonal runtime speedup for A-NHP, discussed near the end of section 3.

---

### Official Review · Reviewer_5rCL · 2021-11-05

**Correctness:** 2
**Technical Novelty And Significance:** 2
**Empirical Novelty And Significance:** 2
**Recommendation:** 5
**Confidence:** 2

**Main Review:**

This paper is clearly the result of detailed study of the of the NDTT modeling framework and its requirements for underlying models. I should start with saying that I'm not an expert in NDTT.

Positive aspects about the paper:
* It discusses clearly and intuitively why the authors made some architectural choices in Sect. 3 and 4.
* Experimental results indicate that the model works.

Challenges for the reader:
* The paper refers to the architecture as a Transformer-like thing, but important differences exist (no feed forward sublayer, use of the bounded tanh as activation function, no layer norms, positional encodings concatted and fed in at each layer). In Sect. 4, "multi-head selective attention" is introduced, which has no relation to standard "multi-head attention" besides the fact that several subnetworks do different things in parallel - even the aggregation of results is different. These differences are not called out explicitly, nor empirically validated.
* Section 5 is extremely dense and very hard to follow, lacking motivation:
  - If you are trying to introduce NDTT, you should provide an example that helps the reader to understand what you are trying to achieve, BEFORE introducing the formalism.
  - $[h]^{(\ell)}(t)$ is used in Eq(9), visually hard to distinguish from $[[ h ]]^{(\ell)}(t)$, and only discussed 10 lines later.
* Experiments: no ablations of the architecture are studied, even though many choices were made that are non-obvious.

Detailed questions:
* Sect. 4: why is the embedding of the event type not sufficient to drive different attention patterns? (i.e., why does the event type need to be part of parameters rather than part of data)

Update after rebuttal
---------------------------
While the authors provided extensive answers to my questions, the updated submission largely does not reflect these answers, nor is it obvious how it could (given the space constraints). Overall, I think the manuscript is not ready for publication at this time, largely due to a lack of clarity in the writing. I feel unable to judge the core technical contribution, but bad writing alone is sufficient for me to decide not to change my rating.

**Summary Of The Paper:**

A model for sequences of events occurring in the continuous time setting is presented. It uses the self-attention mechanism from the Transformer model, together with some domain-specific masking operations, to create event embeddings that are contextualized by their relation to other (relevant) events that occurred. The model is then extended towards a generative process, which in turn is integrated into a neural datalog through time framework.
Experiments show the model to be perform equally or better than baselines based on RNNs or discrete time embeddings.

**Summary Of The Review:**

Overall, I found the first four pages of this paper quite enjoyable, but found the remainder large impenetrable as someone unfamiliar with the area. Given the fact that obvious ablation experiments are missing, I do not think that this paper is ready for publication, lacking both in clarity of writing and strength of evaluation.

---

> ### Author Response · Authors · 2021-11-23
> **ablation issues**
>
> Thank you for the careful read and for raising questions about our experimental design.
>
> > The paper refers to the architecture as a Transformer-like thing, but important differences exist (no feed forward sublayer, use of the bounded tanh as activation function, no layer norms, positional encodings concatted and fed in at each layer). [...]
>
> > Experiments: no ablations of the architecture are studied, even though many choices were made that are non-obvious.
>
> But what we report here was actually an ablation.  We disclosed this in appendix C.3: “In our preliminary experiments, we also tried adding feed-forward layers and layer normalization (Ba et al., 2016) as in the original Transformer architecture (Vaswani et al., 2017), but they didn’t help.”
>
> Since the simpler architecture worked as well, we presented only this simpler architecture in the main text, in order to keep section 2 concise and keep equations (1)-(4) simple for the reader.
>
> The design choices of concatenating the positional encodings (rather than adding them) and feeding them in at higher layers were inspired by previous work (Kitaev & Klein 2018 and He et al. 2020).  As in those papers, we hoped that this would better expose the positional information.  After all, the stacked LSTM approach in prior work was able to directly see the passage of time at every layer, so we figured that the Transformer should be allowed the same.  See also our reply to R4 (reviewer 9eGx).
>
> We’d be happy to discuss these choices further in an appendix.  Should we have tested them empirically?  We could, but it’s not clear that this makes the paper more interesting or useful.  Our goal was to show that “attention is all you need” for NHP and NDTT: i.e., that a Transformer architecture -- by which we mean layered attention without any other sequential state -- can be designed for the NHP and NDTT frameworks and successfully used in place of an LSTM architecture.  We wrote down an approach that we found to be simple and intuitive, and found experimentally that it worked well enough to make our point.
>
> The original Transformer paper similarly had the goal of showing that attention is all you need.  It made some sensible initial choices and didn’t explore every possibility.  Dozens of subsequent publications then presented variant Transformer architectures that they claimed to be better on some dataset (or more efficient).
>
> The specific new ingredient in our paper is to model the intensity of a possible event by attention on irregularly spaced contextual events, where the attention can be controlled by user-written rules.
>
> Obviously this new ingredient can be mixed and matched with lots of other small but potentially important architectural choices. You may think that we should have done a full architecture search to identify the very best combination.  But the best combination may depend on domain, so there is no guarantee that the argmax would generalize beyond our particular experimental datasets.
>
> Anyone who wants to apply the method in a Really Important Domain had better do their own architecture search.  Our contribution is to enable their search to include Transformers as well as LSTMs!  They are certainly free to vary the Transformer details.

---

> ### Author Response · Authors · 2021-11-23
> **multi-head selective attention**
>
> > In Sect. 4, "multi-head selective attention" is introduced, which has no relation to standard "multi-head attention" besides the fact that subnetworks do different things in parallel -
>
> No relation?  The analogy is quite tight.  The traditional idea of having multiple heads is that each head looks at the context through its own set of eyes -- that is, its own attention weights and value projections, as determined by its own parameters.  And that’s exactly what we’re doing.  Each rule looks at the context through its own attention weights determined by its own parameters.  The rule already specifies symbolically which past events can get attention > 0 in the first place, so it makes sense for the rule to also provide the parameters that determine the attention weights and value projections.
>
> > even the aggregation of results is different.
>
> You evidently mean that our equations (7) and (9) sum the heads rather than projecting a concatenation of the heads.  But that’s mainly a presentational difference.  Vaswani et al. (2018, page 5) write:
>      Multihead(Q,K,V) = Concat(head_1,...,head_h) W^O
>      where head_i = Attention(QW^Q_i, KW^K_i, VW^V_i)
>      and W^O has dimensions h d_v \times d_model.
>
> We can therefore express
>      W^O = Stack(W^O_1, …, W^O_h)
>      where each W^O_i has dimension d_v \times d_model.
>
> Then
>     Multihead(Q,K,V) = \sum_i head_i W^O_i = \sum_i head’_i
>     where head’_i = Attention(QW^Q_i, KW^K_i, VW’^V_i) for W’^V_i = W^V_i W^O_i.
>
> So their approach really is equivalent to our approach of summing over the heads, if the value projection matrix for head i is changed from W^V_i to W’^V_i.
>
> The only substantive difference is that we learn each W’^V_i directly as a matrix with dimensions (d_model \times d_model), whereas Vaswani et al. in effect learn it as a product W^V_i W^O_i with dimensions (d_model \times d_v) and (d_v \times d_model), ensuring that it has rank <= d_v.
>
> In the case where d_v = d_model, the two methods are equivalent (but ours is more efficient since it has half as many parameters and saves a matrix multiplication).
>
> And we deliberately chose d_v = d_model because in our setting, most fact tokens are only influenced by one or two rule types i -- that is, for most embeddings, most heads are actually inactive due to our structural zeroes.  Thus, we wanted to allow each rule that did affect the fact’s embedding to be able to contribute to all of the dimensions of the fact rather than just a few.  Furthermore, it was feasible for us to learn a d_model \times d_model matrix for each rule since our d_model was rather small, typically 32 or 64 (a database fact usually doesn’t need a very high-dimensional embedding).
>
> Note that even for d_v < d_model, we could still recover Vaswani et al.’s method by learning our W’^V_i under a rank constraint, for example by explicitly learning it as a product.
>
> > Sect. 4: why is the embedding of the event type not sufficient to drive different attention patterns? (i.e., why does the event type need to be part of parameters rather than part of data)
>
> We want to allow the domain designer to specify structural zeroes in the attention pattern.  Section 4 is a warmup for section 5, which has a pattern-matching language that allows specifying the relationships among millions of event types.  Most of these event types don't influence one another directly.  The domain designer can write that down.
>
> You are correct that with a large amount of training data, the model could learn that certain kinds of attention weights are approximately 0.  But why learn that from data if it's a reasonable domain assumption?  (In the same way, a fully connected graphical model could learn conditional independencies with enough training data, but if they are a reasonable domain assumption, you should enforce it by leaving certain edges out of the graphical model.)

---

> ### Author Response · Authors · 2021-11-23
> **presentation of work**
>
> > Experiments show the model to perform equally or better than baselines based on RNNs or discrete-time embeddings.
>
> All the baselines we considered use continuous-time embeddings.
>
> > Overall, I found the first four pages of this paper quite enjoyable, but found the remainder large impenetrable as someone unfamiliar with the area.
>
> Thanks.  As far as we can tell, your difficulty was with section 5.  We have revised it.  Better?
>
> > If you are trying to introduce NDTT, you should provide an example that helps the reader to understand what you are trying to achieve, BEFORE introducing the formalism.
>
> Paragraphs 3-4 of the introduction did give such an example.  In the rebuttal version, we have continued the discussion of this example in section 5.
>
> (We are not trying to introduce NDTT, though.  NDTT was introduced by Mei et al. (2020), who gave a clear explanation for what they were trying to achieve.  We are trying to synthesize a Transformer that respects the domain-specific influence diagram conveyed by an NDTT program.)

---

### Official Review · Reviewer_vfGf · 2021-11-06

**Correctness:** 3
**Technical Novelty And Significance:** 3
**Empirical Novelty And Significance:** 3
**Recommendation:** 6
**Confidence:** 3

**Main Review:**

While the paper is not too theoretical, the techniques do seem to appear natural and well-motivated by the problem at hand. The experiments are well done and reproducible with the appendix.

I did find the paper a bit difficult to read, but this may be somewhat unavoidable due to the probabilistic modeling. I am not sure about the claim of A-NDTT being simpler and shallower when training time is longer? There is a lot of emphasis on probability models and interpretability with neural-symbolic computing in the exposition that is unexplored in the experiments, too.

**Summary Of The Paper:**

The paper introduces A-NHP and A-NDTT that tackle the problem of continuous time sequences and neural-symbolic computing with a time component using transformers and attention. The paper explains their model and techniques mainly with equations. The work seems primarily motivated by previous work on Hawkes processes and neural datalog through time NDTT. They improve by adding a probability model that allows them to embed all possible events and using transformers instead of LSTMs for parallelization. They achieve strong experimental results, especially in StackOverflow and RoboCup.

**Summary Of The Review:**

I would recommend the paper for acceptance. While there are some issues, the paper is generally well-written, explores new techniques, achieves good results.

---

> ### Author Response · Authors · 2021-11-23
> **presentation and claim of “shallow architecture”**
>
> Thank you for acknowledging our new techniques and good results.
>
> The paper as submitted was indeed quite dense due to page limits.  As noted in the general response, we have expanded the background material in section 5 for you.
>
> The Transformer computation graph really is shallower and broader.  That was an important and deliberate design choice in Vaswani et al. (2017).  It has two benefits.  First, shallowness may improve the optimization landscape.  (That’s why deeper architectures can benefit from including skip connections or their “horizontal” equivalent, LSTM forget gates.)  Second, replacing depth with breadth allows more parallelism.  The increased parallelism is seen empirically in our speedup from NHP to A-NHP (see Table 2 in Appendix C.3), where both are run on GPU.
>
> It’s true that we don’t see this speedup from NDTT to A-NDTT, as you noted.  But that’s only because we only implemented those models on CPU, which doesn’t exploit the parallelism.  In Appendix C.4, we explain how a GPU could be used.
>
> (We also regard the Transformer architecture as simpler, for reasons explained in (3) at the end of section 1.)

---

### Author Response · Authors · 2021-11-23
**presentation and main contribution**

**presentation**

While we think the research is mature, we can see that some reviewers hit a wall when they reached the start of section 5.  We apologize that our presentation of the NDTT formalism was too condensed to follow in the previous version.  We had compressed it rather savagely to fit within the page limits.

Thus, in the rebuttal version, we have substantially expanded the background explanation of NDTT.  These changes are marked by a green changebar in the right margin at the start of section 5.  (We made room by moving some experimental results into an appendix.)  We also eliminated the use of Mei et al.’s “launch/dock” terminology, using the more intuitive “add/remove” instead.

We hope that this addresses your concerns about the presentation.

(Of course, you can get an even better understanding of NDTT by looking at the original paper at http://proceedings.mlr.press/v119/mei20a/mei20a.pdf.)

**what is the main contribution?**

R4 (Reviewer 9eGx) asks about our main contribution. Our paper is addressed to readers who are already familiar with previous work on neural temporal point processes and are interested in how to replace LSTMs with Transformers throughout that line of work.  Switching to Transformers is a natural idea to consider, for reasons explained in the introduction.  Our full approach shows how to accomplish this redesign for the fancy NDTT framework of Mei et al. (2020).

But NDTT = NHP + domain knowledge. So we showed first how to modify NHP to use Transformers.  Then we showed how to generalize this modification to NDTT. The NHP version (section 3) is simpler, so it was pedagogically useful to present first, and it is useful on its own. The resulting model, A-NHP, outperforms other Transformer-based point processes (Zuo et al. 2020, Zhang et al. 2020), perhaps because our novel continuous-time Transformer architecture allows the attention weights to be time-varying between events.

As the reviewer says, NDTT is indeed “a neuro-symbolic framework for embedding rules of which event can draw information from which other events.”  Those rules come from prior work (Mei et al. 2020). What we proposed is a modification to the "neuro" part, leaving the "symbolic" part alone.

Our new version of “which event can draw information from which other events” is a nontrivial change because both information flow and the passage of time are handled differently in Transformers than in the LSTM past work:

* Prior work (NDTT): An event immediately updates the LSTM cells of relevant current facts, which then drift over time.
* Our contribution (A-NDTT): A current fact’s temporal embedding at each layer uses attention to look back at relevant past events and their temporal embeddings.

We will resolve other individual concerns in our responses to individual reviews.

---

### Decision · Program_Chairs · 2022-01-20

**Decision:**

Accept (Poster)

**Comment:**

The paper builds upon previous work on neural temporal point processes. It mainly proposes the replacement of the LSTMs with Transformers as transformers are widely considered as a more powerful sequence modeling tool and the three advantages listed in the end of section 1 in this paper.

However, on the modeling side, it is not straight-forward how to apply the transformer (the attention architecture) on to the continuous time-sequence problem using NDTT. I think because I read a revised version of this paper, it is actually more understandable to me as compared to the reviewers who read the first draft of the paper. I think A-NDTT is a natural and principled way of introducing the attention mechanism into the continuous time neural symbolic framework, although I agree it unfortunately does not leading to a significant improvement in every experimental setting.

To summarize the discussions, I think the authors did a good job in resolving the concerns on the related work and made the paper easier to understand. I appreciate these efforts from the authors even though I also understand there are concerns left still from the reviewers.

In summary, I am leaning to accept this paper because I think it is an interesting contribution. However, I do agree with the reviewers that the writing of the paper needs to be improved and the experiment section is relatively weak in this paper.

---

> ### Public Comment · ~Jason_Eisner1 · 2022-03-21
> **rewrites in final version**
>
> Thanks very much for judging this paper to be sufficiently interesting to be above threshold!
>
> For the camera-ready version submitted last week, we made substantial further rewrites -- particularly overhauling the abstract and sections 1 and 5, which seem to have been the difficult parts.  The paper should stand alone better now as it includes a clearer explanation of the NDTT formalism.  We also added a discussion of various architectural choices and alternatives in Appendix A, including issues that concerned some of the reviewers.